# SPO11 dimers are sufficient to catalyse DNA double-strand breaks in vitro

Cédric Oger[1] & Corentin Claeys Bouuaert[1✉]

SPO11 initiates meiotic recombination through the induction of programmed DNA double-strand breaks (DSBs)[1,2], but this catalytic activity has never been reconstituted in vitro[3,4]. Here, using *Mus musculus* SPO11, we report a biochemical system that recapitulates all the hallmarks of meiotic DSB formation. We show that SPO11 catalyses break formation in the absence of any partners and remains covalently attached to the 5′ broken strands. We find that target site selection by SPO11 is influenced by the sequence, bendability and topology of the DNA substrate, and provide evidence that SPO11 can reseal single-strand DNA breaks. In addition, we show that SPO11 is monomeric in solution and that cleavage requires dimerization for the reconstitution of two hybrid active sites. SPO11 and its partner TOP6BL form a 1:1 complex that catalyses DNA cleavage with an activity similar to that of SPO11 alone. However, this complex binds DNA ends with higher affinity, suggesting a potential role after cleavage. We propose a model in which additional partners of SPO11 required for DSB formation in vivo assemble biomolecular condensates that recruit SPO11–TOP6BL, enabling dimerization and cleavage. Our work establishes SPO11 dimerization as the fundamental mechanism that controls the induction of meiotic DSBs.

Despite it being more than 25 years since the discovery of SPO11 as the DNA-cleavage subunit that initiates meiotic recombination[1,2], SPO11 activity has long been recalcitrant to biochemical reconstitution, thereby hindering a detailed understanding of the mechanism and regulation of meiotic double-strand break (DSB) formation.

SPO11 evolved from the DNA-cleavage (A) subunit of a heterotetrameric (A$_2$B$_2$) type IIB topoisomerase, topoisomerase VI (topo VI)[2,5] Like topo VI, SPO11 cleaves DNA by a pair of nucleophilic attacks from an active site tyrosine to the DNA backbone, producing breaks with two-nucleotide (nt) 5′ overhangs with the transesterase covalently attached to the 5′ DNA ends[6–8]. Each DNA strand is cleaved by a hybrid active site comprising the catalytic tyrosine from one subunit, with a metal ion-binding pocket contributed by the second[9,10]. Whereas topo VI modulates DNA topology by cycles of gate opening (DNA cleavage), strand passage and gate closing, orchestrated by the ATP-dependent topo VIB subunit[11], SPO11, by contrast, cleaves DNA without restoration of the broken strands. Nevertheless, in vivo, the cleavage activity of SPO11 depends on a topo VIB-like subunit (TOP6BL) and a series of additional accessory factors[12]. In mouse, these include meiosis-specific partners REC114, MEI4, MEI1 and IHO1 (RMMI)[13–16]. However, in the absence of a reconstituted system, the mechanism that controls SPO11 activity and the function of the partners remains unknown.

Here we show that mouse SPO11 has intrinsic DNA-cleavage activity in vitro and explore the factors that impact target site selection and cleavage. We find that SPO11 cleavage is inherently limited by its monomeric state, and propose that SPO11 dimerization constitutes the fundamental mechanism that controls the initiation of meiotic recombination.

## Mouse SPO11 cleaves DNA in vitro

To set up an in vitro system to study meiotic DSB formation, we purified maltose-binding protein (MBP)-tagged *M. musculus* SPO11 from baculovirus-infected insect cells (Fig. 1a,b). We incubated protein fractions from an ion exchange column in the presence of plasmid DNA and divalent metal ions, stopped reactions after 2 h with EDTA and SDS and deproteinized samples with proteinase K before agarose gel electrophoresis. The peak of SPO11 coincided with robust DNA cleavage activity, producing single-strand (nicked), double-strand (linear) and multiple DSBs on the supercoiled substrate (Fig. 1c,d). This cleavage activity was abolished when the tandem active-site tyrosine residues (Y137 and Y138) found in SPO11 were mutated to phenylalanine (Y137F/Y138F) (Fig. 1e). In addition, cleavage was dependent on the presence of a divalent metal ion, with Mn$^{2+}$ being more effective than Mg$^{2+}$, and Ca$^{2+}$ also supporting low levels of cleavage (Fig. 1f and Extended Data Fig. 1). Hence, mouse SPO11 has intrinsic double-strand DNA cleavage activity, even in the absence of any of the partners known to be required in vivo.

## SPO11 binds covalently to 5′ DNA ends

We verified that SPO11 remains covalently attached to DNA breaks in four different ways (Fig. 2a).

First, we asked whether the linear cleavage product was dependent on deproteination of the sample before electrophoresis. Indeed, in the absence of proteinase K, the linear product was not detected and was instead converted to a smear due to the presence of covalently bound denatured SPO11 proteins (Fig. 2b).

[1]Louvain Institute of Biomolecular Science and Technology, Université Catholique de Louvain, Louvain-La-Neuve, Belgium. ✉e-mail: corentin.claeys@uclouvain.be

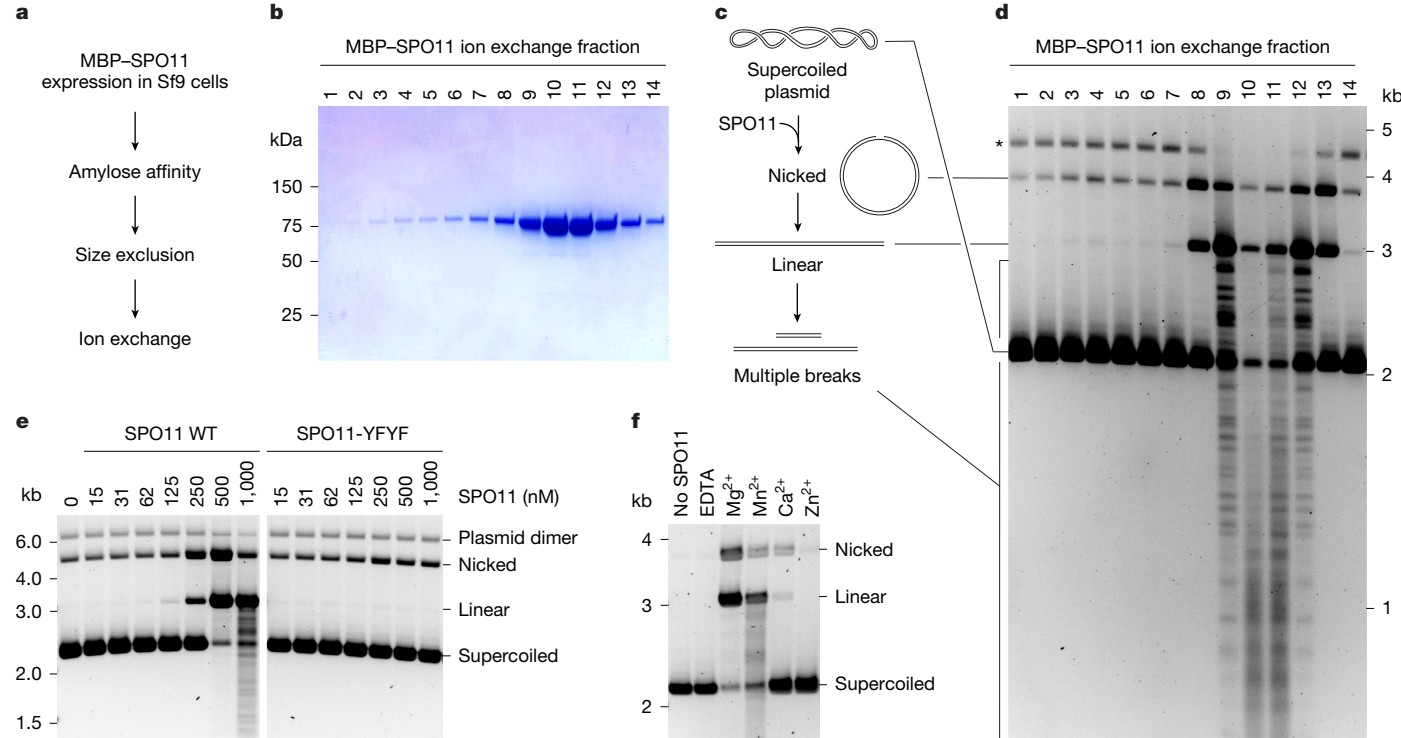

**Fig. 1 | In vitro reconstitution of SPO11-dependent DNA DSB formation.**
**a**, Purification scheme of mouse SPO11 protein. **b**, SDS–PAGE analysis of ion
exchange fractions of purified MBP–SPO11. **c**, Scheme of the in vitro DNA
cleavage assay. Products are illustrated following deproteination of samples
with proteinase K. **d**, Plasmid DNA cleavage analysis using fractions of SPO11
from **b** in the presence of divalent metal ions (Mg²⁺ and Mn²⁺). The band labelled
by an asterisk corresponds to a plasmid dimer. **e**, Effect of active site-directed
double mutation Y137F/Y138F (YFYF/) on the DNA cleavage activity of SPO11.
WT, wild type. **f**, Requirement for divalent metal ions in DNA cleavage activity
of SPO11. For gel source data, see Supplementary Fig. 1.

Second, we tested whether phenol–chloroform extraction would
lead to the enrichment of covalent protein–DNA adducts in the organic
phase. As expected, cleaved DNA was detected in the organic phase in
the presence of wild-type SPO11, but not the Y137F/Y138F double
mutant (Fig. 2c). When samples were treated with proteinase K before
phenol extraction, cleavage products were absent from the organic
phase and were instead found in the aqueous phase.

Third, we asked whether broken DNA ends were protected from
degradation by a 5′–3′ exonuclease. Whereas a linear product gen-
erated by digestion with EcoRI was degraded in the presence of T5
exonuclease, SPO11 cleavage products were resistant to exonuclease
treatment (Fig. 2d).

Fourth, we performed cleavage reactions with 5′ or 3′ fluorescently
labelled 80-bp duplex DNA substrates, and separated covalent com-
plexes by SDS–PAGE. A fluorescent signal of slightly lower electro-
phoretic mobility than MBP–SPO11 was detected with the 3′-labelled
substrate, but not the 5′-labelled substrate, and was absent with the
Y137F/Y138F mutant (Fig. 2e). Hence, following cleavage, SPO11 remains
covalently bound to 5′ broken DNA strands, as previously observed
in vivo[6,7].

## Cleavage sites and substrate preferences

To establish whether SPO11 produces the expected staggered breaks
with two-nucleotide 5′ overhangs, we performed cleavage reactions
with 5′-labelled 80-bp duplexes and separated cleavage products
by denaturing gel electrophoresis. We detected sites of preferential
cleavage, creating products of 16, 30, 48 and 63 nucleotides on the top
strand and 15, 30, 48 and 62 nucleotides on the bottom strand (Fig. 3a).
The position of the cleavage sites confirms the staggered cleavage
pattern of SPO11 (Fig. 3a, bottom). Seven out of the eight preferred
cleavage sites had a guanosine in position −3 with respect to the dyad

axis, suggesting that base composition of the substrate influences
SPO11 activity.

In mice, the distribution of SPO11-dependent breaks across the
genome is largely determined by sequence-specific binding of the
H3K4 methyltransferase PRDM9, which defines the position of around
100–300-bp regions of increased DSB activity (known as DSB hot-
spots)[17–20]. However, whether intrinsic substrate preferences of mouse
SPO11 contribute to fine-scale selection of cleavage sites within hot-
spots is unknown.

In addition to potential contacts with DNA bases, substrate selec-
tion by SPO11 could be influenced by structural features of the double
helix—for example, DNA bending or unwinding. To explore the factors
that impact SPO11 target site selection, we digested the products of our
standard plasmid cleavage reaction with a restriction endonuclease.
Agarose gel analysis showed a non-random distribution of cleavage
sites across the plasmid (Fig. 3b). A single prominent cleavage site
mapped within a synthetic sequence cloned within the pUC-derived
plasmid, whereas cleavage was much less efficient within the plasmid
backbone. Using a DNA bendability prediction tool, DNAcycP[21], we
found that the preferential cleavage site corresponded to a peak of
predicted DNA bendability (Fig. 3b, bottom right).

To test whether DNA bendability affects SPO11 target site selec-
tion, we performed cleavage reactions using a plasmid containing
24 repeats of the highly bendable Widom 601 sequence, a strong
nucleosome-binding sequence[22]. Restriction digestion of SPO11 cleav-
age reactions produced a markedly periodic pattern, indicating that
Widom 601 sequences produce hotspots for DNA cleavage (Fig. 3c).
To confirm this, we cloned one, three and six copies of the Widom 601
sequence into a pUC19 plasmid and used this as a PCR template to cre-
ate linear substrates with a fluorophore at one extremity. Agarose gel
analysis of SPO11 cleavage reactions allows us to map cleavage prod-
ucts along the linear substrate, which shows a correlation between the

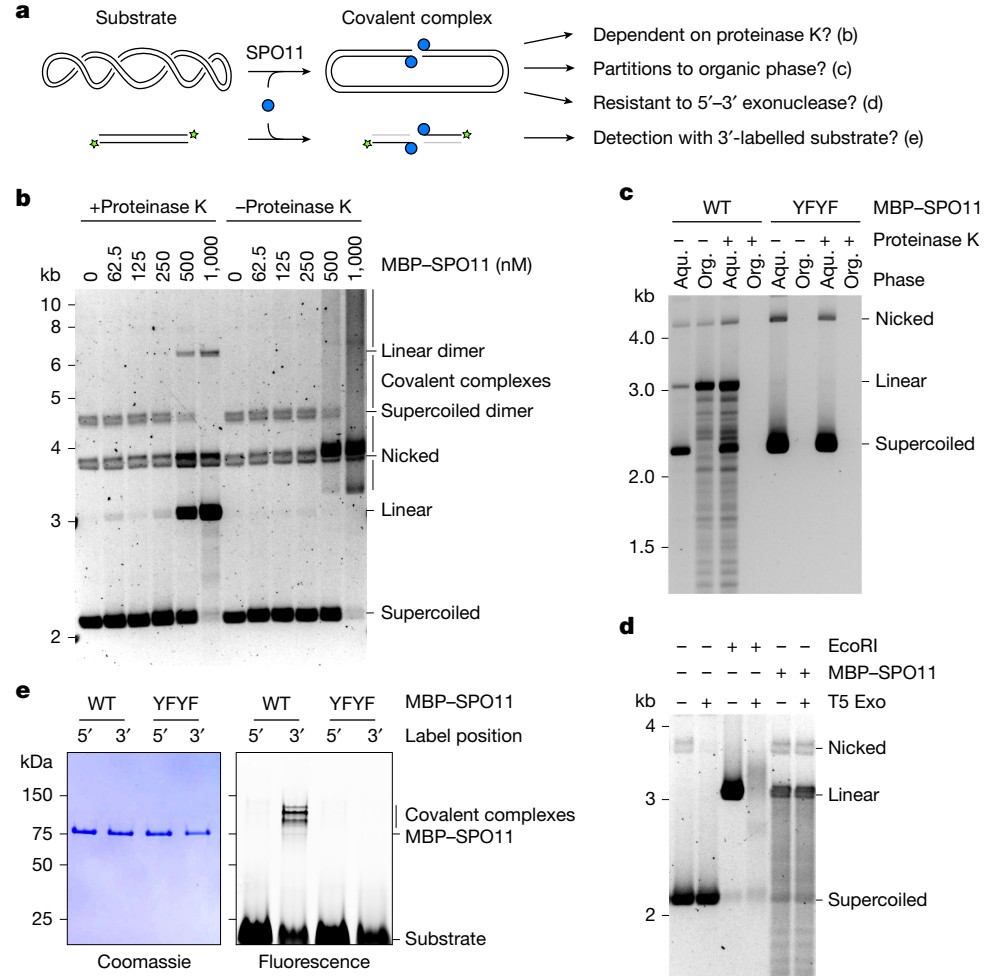

**Fig. 2 | SPO11 is covalently bound to 5′ DNA ends. a**, Predictions tested in **b**–**e**. **b**, Agarose gel analysis of DNA cleavage products with or without proteinase K treatment before electrophoresis. All reactions were treated with SDS to eliminate non-covalent binding. **c**, Phenol–chloroform partitioning of DNA cleavage products with wild-type and mutant SPO11, with or without proteinase K treatment before phenol extraction. **d**, Analysis of the resistance of EcoRI- and

SPO11-dependent cleavage products to the 5′–3′ exonuclease T5 Exo. **e**, SDS–PAGE analysis of covalent SPO11–DNA complexes from cleavage assays using wild-type or mutant SPO11 (YFYF) in the presence of 3′ or 5′ fluorescently labelled 80 bp substrates. For gel source data, see Supplementary Fig. 1. Aqu., aqueous phase; Org., organic phase.

cleavage sites, the position of Widom 601 sequences and the predicted bendability of the substrate (Extended Data Fig. 2). Nevertheless, the peak of DNA bendability did not align with the preferred cleavage site, and some sites predicted to be bendable did not produce strong cleavage sites, indicating that bendability alone as predicted by DNAcycP is not sufficient to account for the target site preference of SPO11. In addition, we cannot exclude that the Widom 601 sequence produces hotspots because of favourable base-specific contacts with SPO11, irrespective of bendability.

Next, we asked whether SPO11 cleavage is affected by the topology of the DNA substrate. Time-course analysis showed that supercoiled plasmids are cleaved more efficiently than nicked plasmids (Fig. 3d). Supercoiled plasmids containing the Widom 601 sequence were cleaved at the same rate as plasmids lacking the Widom sequence. By contrast, the presence of Widom sequences accelerated the cleavage of linear plasmids, suggesting that supercoiling promotes cleavage by facilitation of DNA bending (Extended Data Fig. 3). Hence, the cleavage activity of SPO11 is probably affected by a combination of inter-related factors including DNA sequence, bending and topology.

Using AlphaFold 3 (ref. 23), we modelled the structure of a SPO11 dimer bound to a duplex DNA substrate (Fig. 3e and Extended Data Fig. 4a,b). The model shows SPO11 poised for catalysis, with the active

site tyrosine placed 3 Å from the correct phosphate groups to produce a break with two-nucleotide 5′ overhangs. The substrate is bent at an angle of 100° with underwound DNA strands at the centre of the complex, consistent with the observed preference of SPO11 for bendable and negatively supercoiled DNA.

## Cleavage requires hybrid active sites

All type II topoisomerases, including SPO11, are thought to cleave DNA using two composite active sites at the interface between two subunits[24–26]. The winged-helix domain of one subunit contributes the catalytic tyrosine, with the Toprim domain of the other subunit contributing metal ion-binding residues (Fig. 4a).

Most SPO11 and Top6A homologues have two tandem tyrosines (Y137 and Y138 in mouse), except for yeast, which has a phenylalanine in the first position (Extended Data Fig. 4c). We confirmed by mutagenesis that Y138 is indeed the catalytic tyrosine in mouse SPO11 (Extended Data Fig. 4d).

The cleavage mechanism proposed for topoisomerases involves two metal ions, in which metal ion A has a direct role in catalysis and metal ion B has a structural role in stabilization of protein–DNA interactions[27,28] (Extended Data Fig. 4e). The hybrid active site modelled by

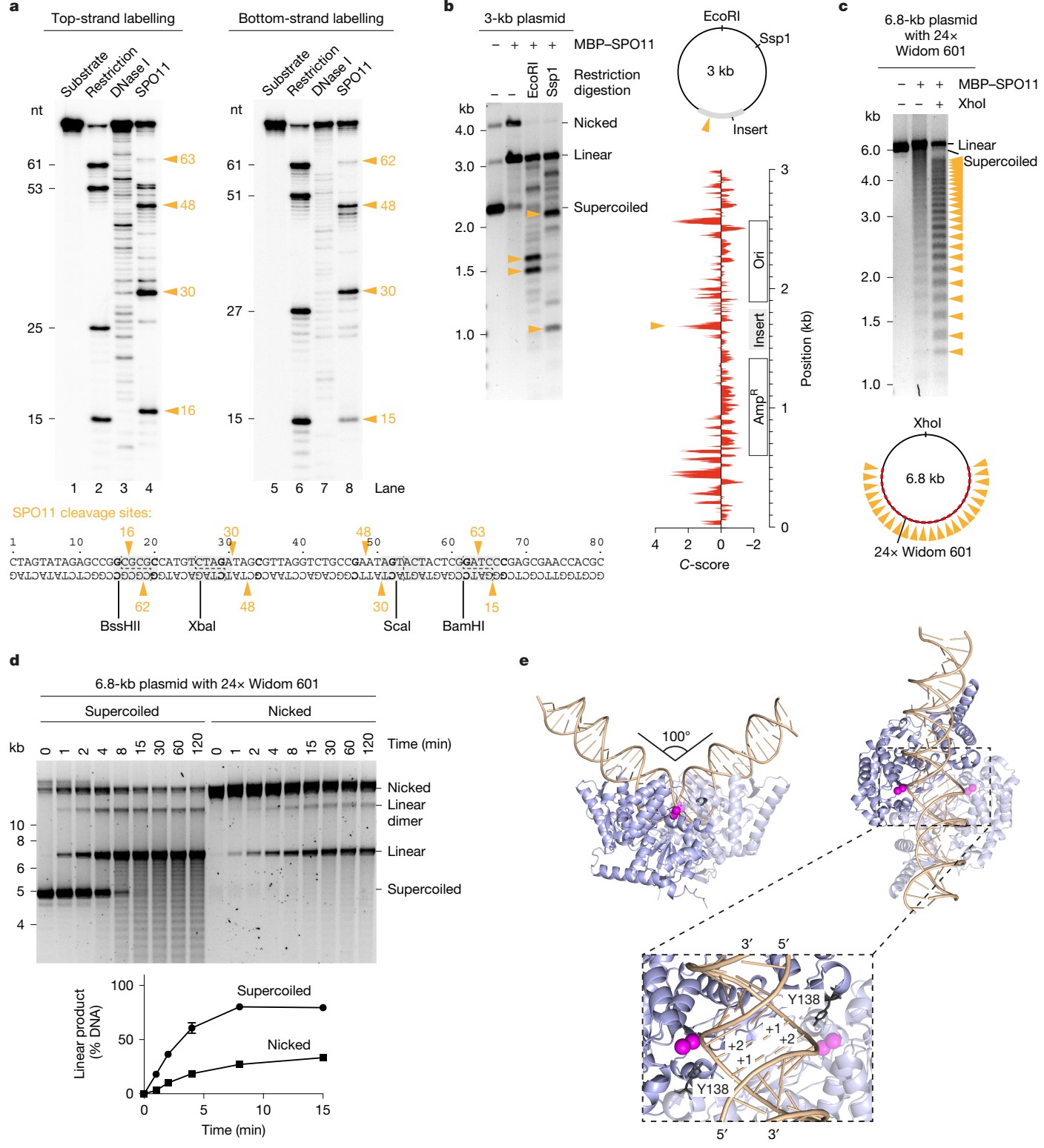

**Fig. 3 | Cleavage pattern and substrate specificity. a**, Sequencing gel analysis of DNA cleavage reactions using 5′ radioactively labelled 80-bp substrates. Lanes 2 and 6 were produced by digestion of the substrate using restriction enzymes indicated below the gel; lanes 3 and 7 were produced by partial digestion of the substrate with DNase I. SPO11 cleavage sites (lanes 4 and 8) are highlighted with orange arrowheads. Positions ±3 from the dyad axes are shown in bold. **b**, Agarose gel analysis of SPO11 cleavage sites on the standard plasmid substrate (pCCB959) using restriction digestion of SPO11 reaction products. Bottom right, cyclizability (*C*-score) of the plasmid substrate,

as predicted by DNAcycP[21]. The positions of the preferential cleavage site are indicated (arrowheads). **c**, Analysis of SPO11 cleavage sites with a plasmid substrate containing 24 copies of the Widom 601 sequence (pOC157). **d**, Effect of DNA topology on the rate of SPO11-dependent cleavage. Quantifications show the mean and range from two independent experiments. **e**, AlphaFold 3 model of SPO11 dimer bound to a 40-bp duplex DNA substrate. Mg²⁺ ions are shown in magenta. The nucleotides that would form the 5′ overhang are labelled +1 and +2. For gel source data, see Supplementary Fig. 1.

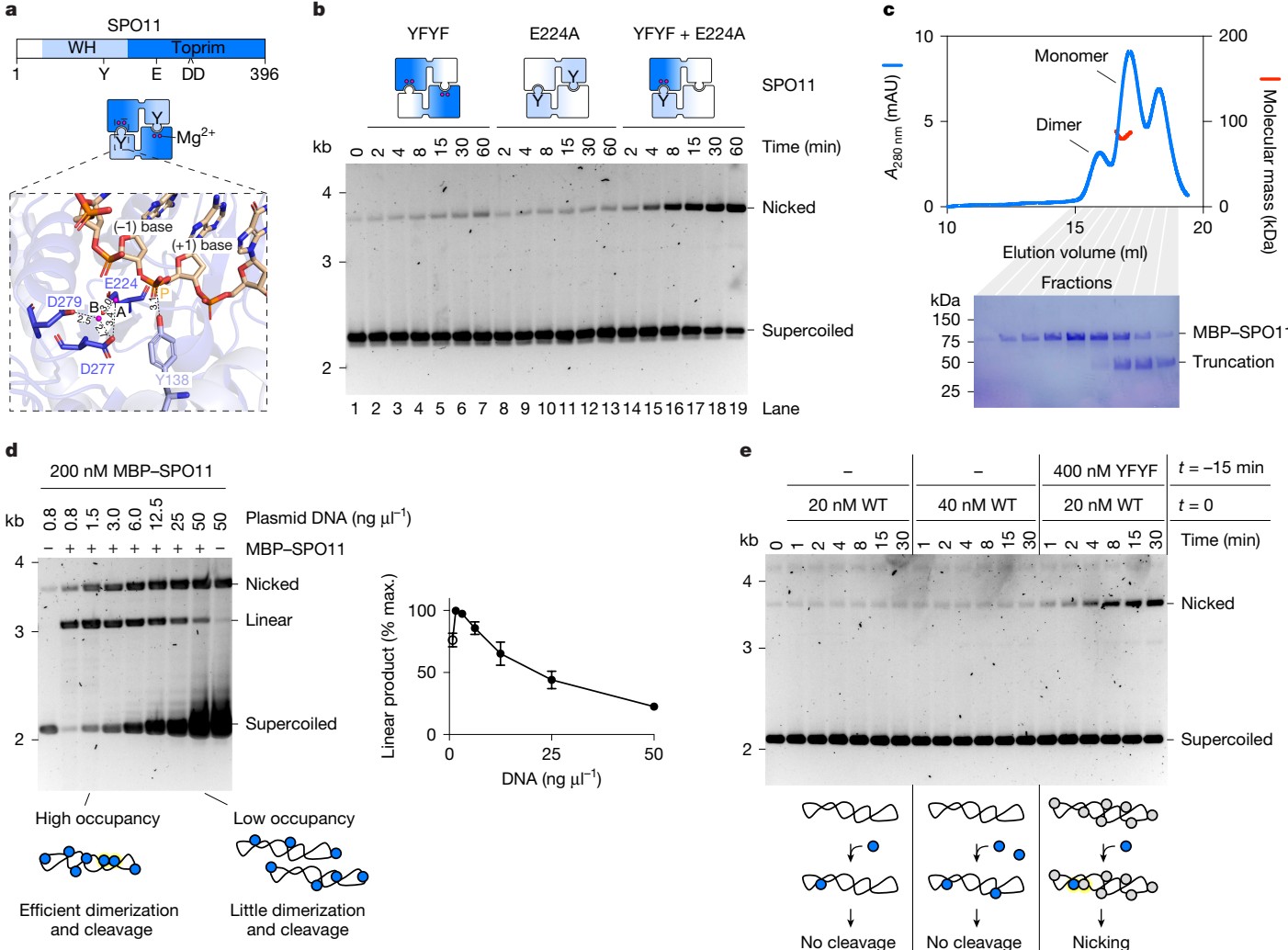

**Fig. 4 | SPO11 cleavage requires dimerization. a**, Domain structure of SPO11 (top) and arrangement of the SPO11 dimer (middle). Bottom, zoomed-in view of the composite active site within the AlphaFold 3 model of a DNA-bound SPO11 dimer. $Mg^{2+}$ ions are labelled A and B. Active site residues and scissile phosphate (P) are shown. **b**, Time-course analysis of SPO11 cleavage with mixtures of two catalytically inactive mutants. **c**, SEC–MALS analysis of MBP–SPO11 following amylose affinity purification. Blue traces represent absorbance measurements at 280 nm derived from size-exclusion chromatography; red traces represent molecular mass measurements across the peak. SDS–PAGE analyses of the corresponding fractions are shown. The leftmost peak probably corresponds to a dimer, although the molecular mass could not be determined; the rightmost peak corresponds to a truncated fragment. **d**, Titration of DNA at a constant

concentration of SPO11; reactions were stopped after 15 min. Quantifications show the mean and range from two independent experiments. At the lowest concentration (open circle), the substrate is limiting so the amount of linear product is not representative of total break levels. **e**, Time-course analysis of SPO11 cleavage at the indicated concentrations of wild-type and catalytically inactive SPO11. With 20 nM wild-type SPO11, the increased nicking activity observed in the presence of the Y137F/Y138F (YFYF) double mutant cannot be explained by doubling the formation of cleavage-competent complexes, because no activity was detected with 40 nM wild-type SPO11. For gel source data, see Supplementary Fig. 1. mAU, milliabsorbance units; max., maximum; WH, winged helix.

AlphaFold 3 places Y138 in proximity to two metal ions, coordinated by residues E224, D277 and D279 (Fig. 4a). These residues are highly conserved in Top6A and SPO11 homologues (Extended Data Fig. 4c), and correspond to the active site residues previously identified in a crystal structure of *Methanocaldoccus jannaschii* Top6A[9]. We infer from sequence alignments that residues E224 and D277 coordinate metal ion A and that residues D277 and D279 coordinate metal ion B. In yeast, E233 and D288 (equivalent to mouse E224 and D277, respectively) are essential for DSB formation, whereas D290 (equivalent to mouse D279) is not[10]. Consistently, we found that mutation of E224 to alanine abolishes the DNA-cleavage activity of mouse SPO11 (Fig. 4b, lanes 8–13).

To formally establish that cleavage involves hybrid active sites, we performed cleavage reactions using mixtures of two catalytically

inactive mutants, Y137F/Y138F and E224A. The assembly of heterodimers should restore one functional active site per dimer, resulting in single-strand DNA breaks. As expected, we found that mixing the two inactive mutants led to the formation of nicked products (Fig. 4b), and hence catalysis requires the concerted action of two SPO11 subunits for assembly of hybrid active sites.

## Dimerization controls DNA cleavage

The mutant protein-mixing experiment suggests either that (1) purified SPO11 is monomeric in solution, as are the yeast Spo11 core complex[29] and *Caenorhabditis elegans* SPO-11 (ref. 30) or (2) dimers undergo rapid subunit exchange. To determine the stoichiometry of mouse SPO11, we subjected 4 µM MBP–SPO11 to size-exclusion chromatography followed

by multiangle light scattering (SEC–MALS) (Fig. 4c). MBP–SPO11 produced a main peak that yielded an experimental molecular mass of $87.25 \pm 1.5$ kDa, consistent with monomeric stoichiometry (expected molecular mass of 87.4 kDa). This peak was preceded by a smaller one that probably corresponds to dimers. Hence, accounting for SPO11 dilution during chromatography, the dissociation constant of SPO11 dimers is probably higher than $1-5\,\mu M$.

The predominantly monomeric stoichiometry of purified SPO11 accounts for its low intrinsic activity. Indeed, although SPO11 is active within a broad range of temperatures (optimum $36-42\,°C$) and pH (6.5–8.5) (Extended Data Fig. 5a,b), cleavage is sensitive to enzyme and substrate concentrations and requires a large excess of protein. For instance, with 12.5 nM (25 ng $\mu l^{-1}$) plasmid, no activity was detected with 60 nM SPO11, and 500 nM SPO11 was required to reach full conversion of supercoiled substrate into linear product within 2 h (Extended Data Fig. 5c). When the substrate concentration was decreased, SPO11 cleaved DNA efficiently at concentrations as low as 50 nM, at least 10–100 times lower than the expected dissociation constant ($K_d$) value (Extended Data Fig. 5d, lane 2). Hence, cleavage is not limited by the pool of soluble dimers but is instead a function of the protein:DNA ratio.

We reasoned that SPO11 binding to DNA would in effect increase its local concentration and perhaps allow dimerization directly on the DNA when substrate occupancy is sufficiently high. This would explain why increasing the DNA concentration inhibits cleavage (Fig. 4d), because reducing plasmid occupancy will lower the likelihood of SPO11 dimerization.

If monomers meet on the substrate, DNA binding should be much more effective than cleavage. Indeed, gel shift analysis shows that SPO11 binds DNA at concentrations that do not support cleavage (Extended Data Fig. 6a–c). Instead, at concentrations that do support cleavage, SPO11 binds so abundantly to the substrate that it provides effective protection against DNase I treatment (Extended Data Fig. 6d,e). In addition, we found that cleavage is significantly more sensitive to salt than DNA binding (Extended Data Fig. 6f–h), probably because a mild reduction in plasmid occupancy caused by increasing salt concentration markedly reduces the likelihood of dimerization. Finally, we found that preincubation of plasmid substrates with an excess of inactive SPO11 facilitates cleavage following the addition of low levels of wild-type protein, presumably because decreasing the search space on the substrate increases the rate of dimerization and cleavage (Fig. 4e). This mostly generates nicked products, indicating that cleavage is caused by heterodimerization of wild-type SPO11 with prebound inactive subunits.

Overall, these data indicate that SPO11 monomers bind efficiently to DNA and that, above a certain threshold, monomers meet on the substrate, allowing cleavage.

## SPO11 can reseal single-strand nicks

Under conditions in which SPO11 mostly cleaves a single DNA strand— for example, in reactions containing a mixture of wild-type and double mutant Y137F/Y138F, or Y137F/Y138F and E224A mutants—we observed additional bands migrating between the positions of the supercoiled and nicked products (Extended Data Fig. 7a,b). Phenol–chloroform extraction shows that these products are not covalently bound to SPO11, indicating that they are plasmid topoisomers (Extended Data Fig. 7c). Time-course analysis demonstrates that these topoisomers accumulate more slowly than cleavage products (Extended Data Fig. 7b).

The formation of topoisomers suggests that SPO11 can sometimes religate broken DNA strands. The most likely scenario is that topoisomers are produced by separation of the two subunits when catalysis is stalled at the single-strand break (Extended Data Fig. 7d). This would result in the swivelling of DNA around the intact phosphodiester bond; reassembly of the dimer interface would then provide an opportunity for reversal of the reaction and liberation of SPO11 from the substrate.

## The SPO11–TOP6BL complex

Mouse SPO11 forms a complex with the topoisomerase-derived TOP6BL subunit (Fig. 5a), which is required for DSB formation in vivo[12]. In topo VI, the B subunit coordinates ATP-dependent dimerization of its GHKL domain with DNA cleavage by Top6A to control strand passage[11]. However, the role of TOP6BL in SPO11-dependent cleavage remains unclear.

To gain insight into the SPO11–TOP6BL complex, we used Alpha-Fold 3 to model a 2:2 heterotetramer bound to a 40-bp duplex DNA (Fig. 5b and Extended Data Fig. 4). Similar to the SPO11–DNA complex presented above, the model shows SPO11–TOP6BL engaged with a bent DNA substrate, with the active site residues poised for cleavage. The complex showed high structural similarity to topo VI, with TOP6BL presenting well-defined GHKL and transducer domains (Extended Data Fig. 8). However, Top6B has a helix two-turn helix motif located between the GHKL and transducer domains, and some species have a structured C-terminal extension absent in TOP6BL. In addition, the ATP-binding site of Top6B is predicted to be degenerated in TOP6BL, consistent with previously reported models[12,31].

To investigate the properties of SPO11–TOP6BL complexes, we purified SPO11–TOP6BL complexes fused with MBP and His–Flag tags, respectively from baculovirus-infected insect cells and analysed complexes using SEC–MALS (Fig. 5c). This showed an experimental molecular mass of $150.2 \pm 2.2$ kDa, consistent with a 1:1 complex (expected molecular mass 157 kDa). Hence, just like SPO11, the cleavage activity of the SPO11–TOP6BL complex is likely to be limited by protein dimerization.

As anticipated, a plasmid cleavage assay showed similar activities between SPO11 and SPO11–TOP6BL complexes, although this was context dependent. For instance, under our standard conditions, at low protein concentrations the SPO11–TOP6BL complex was slightly more active than SPO11 alone (Fig. 5d, lanes 2 and 4). However, in the presence of 0.01% NP-40, the cleavage activity of SPO11 was strongly stimulated whereas that of the SPO11–TOP6BL complex was not (lanes 6 and 8). We conclude that the physicochemical parameters of the reaction impact differently the cleavage activity of SPO11 and SPO11–TOP6BL complexes, perhaps due to the effects of TOP6BL on DNA binding. Indeed, purified mouse TOP6BL was recently shown to bind DNA[32]. In addition, the B subunit of topo VI directly contacts DNA to coordinate substrate binding with ATP-dependent dimerization and catalysis[33].

We previously showed that purified yeast Spo11 complexes, despite being catalytically inactive, bind with high affinity to DSBs through non-covalent interactions[29]. To investigate the end-binding activity of mouse homologues, we performed gel shift analyses using a short hairpin substrate with a two-nucleotide 5′ overhang at one extremity, to which the yeast complex binds with subnanomolar affinity. We found that the SPO11–TOP6BL complex binds efficiently to this substrate, whereas SPO11 alone does not (Fig. 5e). This confirms that TOP6BL affects the DNA-binding properties of SPO11 and supports a model in which tight end binding by SPO11–TOP6BL could impact DSB processing, as we previously proposed for the yeast complex[4,29,34].

## Discussion

Here we reconstituted the formation of meiotic DNA DSBs in vitro using mouse SPO11. We demonstrate that the in vitro assay recapitulates all of the features expected of a bona fide SPO11 cleavage reaction: catalysis depends on the catalytic tyrosine Y138 and divalent metal ions; protein remains covalently bound to the 5′ DNA strand; and each strand is cleaved by a composite active site assembled at the interface of two SPO11 monomers, with the two cleavages staggered to produce a two-nucleotide 5′ overhang. In addition, we show that SPO11 cleavage site selection is driven by a mild sequence bias and a preference for bendable and underwound DNA, and that SPO11 is able

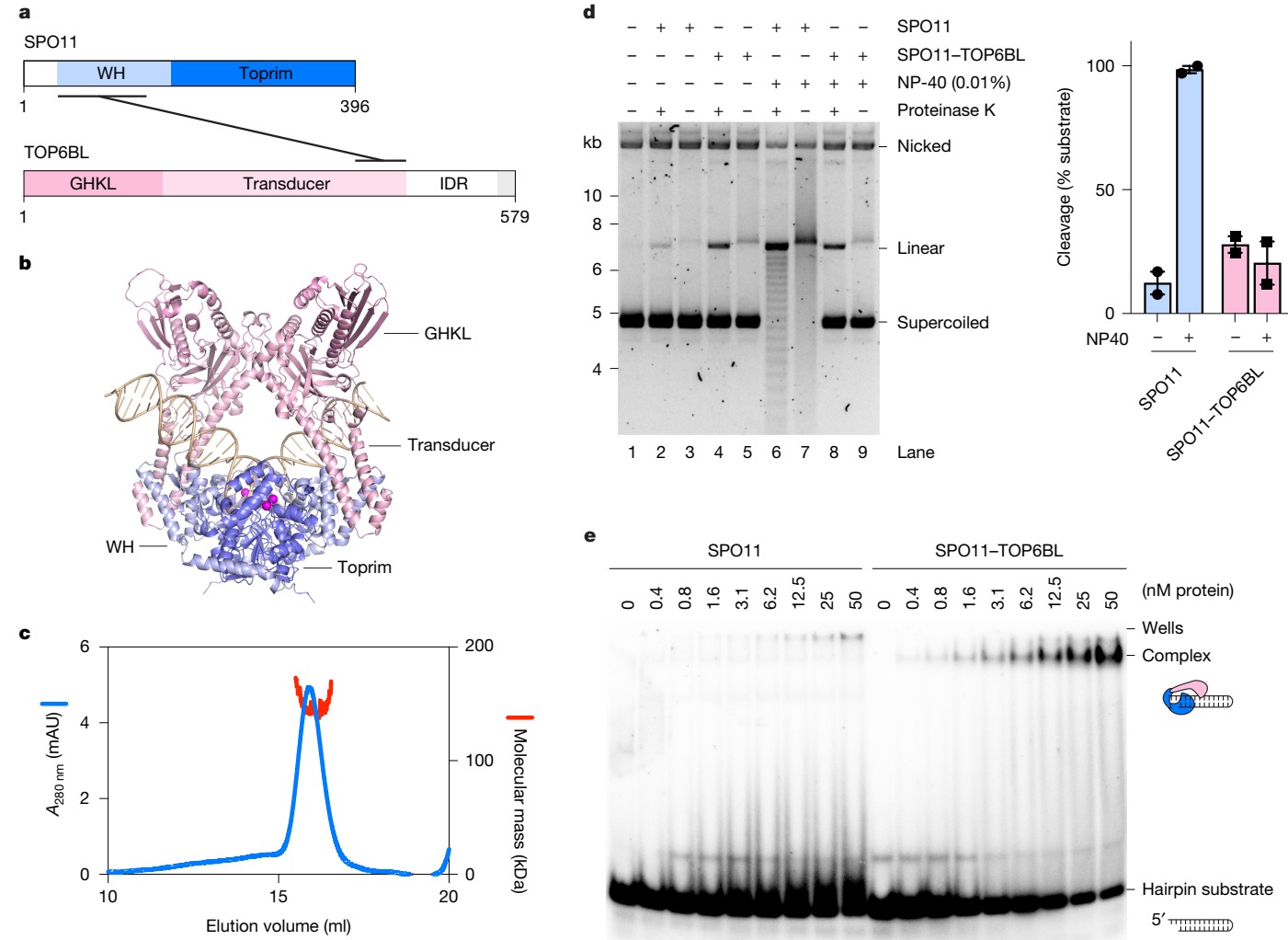

**Fig. 5 | The SPO11–TOP6BL complex. a**, Domain structure of SPO11 and TOP6BL. The C terminus of TOP6BL (grey) binds REC114 (ref. 31). **b**, AlphaFold 3 model of the SPO11–TOP6BL heterotetramer bound to a 40-bp DNA substrate. The TOP6BL intrinsically disordered region (IDR) was omitted from the model. **c**, SEC–MALS analysis of SPO11–TOP6BL complexes tagged with MBP and His–Flag, respectively. Blue traces represent absorbance measurements at 280 nm from SEC; red traces represent molecular mass measurements across the peak. **d**, Plasmid (pOC157) cleavage with 45 nM SPO11 or SPO11–TOP6BL complexes with or without 0.01% NP-40. In lanes 3, 5, 7 and 9, proteinase K treatment was omitted before electrophoresis. Quantifications show individual data points, mean and range from two independent experiments. **e**, Gel shift analysis of the binding of SPO11 and SPO11–TOP6BL complexes to a 25-bp hairpin substrate with a two-nucleotide 5′ overhang. For gel source data, see Supplementary Fig. 1.

to reseal a broken strand when stuck in a single-strand nicked intermediate. SPO11 cleavage is inherently controlled by its monomeric state and occurs in vitro following dimerization on the DNA substrate. The two accompanying papers[35,36] report similar findings focusing on the SPO11–TOP6BL complex.

Our results provide a framework with which to understand the mechanism and control of meiotic DNA DSB formation. First, although SPO11 cleaves DNA independently in vitro, we suggest that its weak dimer interface renders SPO11 dependent on accessory factors in vivo (Extended Data Fig. 9). Because the yeast DSB machinery is assembled through a DNA-dependent condensation mechanism[37], we propose that partitioning of SPO11–TOP6BL complexes within RMMI condensates allows SPO11 to reach a critical threshold required for dimerization. This would facilitate precise spatiotemporal control of DSB formation (Supplementary Discussion 1).

Second, the influence of DNA sequence and topology on SPO11 cleavage suggests that these factors influence the fine-scale DSB landscape in mice. In addition, AlphaFold modelling suggests that SPO11 complexes across eukaryotes bend the DNA substrate before cleavage (Extended Data Fig. 10). This could lead to preferential DSB

induction at sites under topological stress, and potentially provide the energy required to drive the cleavage reaction forward (Supplementary Discussion 2).

Third, in addition to the expected DSB activity, we found that SPO11 exhibits DNA-nicking activity. This indicates that cleavage of the two DNA strands is not strictly coordinated and that the SPO11 dimer may collapse during catalysis (Supplementary Discussion 3).

Fourth, reaction conditions that promote the accumulation of single-strand cleavage products tend to yield covalently closed topoisomers, probably through a swivelling mechanism. This may also result from the instability of SPO11 dimers (Supplementary Discussion 4).

Fifth, plasmid relaxation activity shows that DNA cleavage is reversible, which is expected because the cleavage reaction is isoenergetic (Supplementary Discussion 5). Hence, DNA breaks can presumably religate until the dimer collapses. The transition from a reversible to an irreversible break is probably accompanied by a conformational change that stabilizes monomeric SPO11–TOP6BL complexes on the DSB (Supplementary Discussion 6). This postcleavage complex would be incompatible with dimerization, thereby promoting the separation of the two DNA ends. Nevertheless, these ends may remain in proximity

through the anchoring of SPO11–TOP6BL to RMMI condensates and loading of MRN complexes and their end-tethering activities[37,38].

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

## Methods

### Preparation of expression vectors

Plasmids are listed in Supplementary Table 1, oligo sequences in Supplementary Table 2 and gBlocks (Integrated DNA Technologies) in Supplementary Table 3.

Sequences coding for *M. musculus* SPO11 and TOP6BL were codon optimized for expression in Sf9 cells and synthesized as gBlocks. The SPO11 gBlock was cloned into pFastBac1-MBP to yield pCCB630 (MBP–SPO11), and the TOP6BL gBlock was cloned into pFastBac1-HisFlag to yield pCCB628 (His–mTOP6BL–Flag).

The SPO11-Y137F/Y138F mutant was generated by QuikChange mutagenesis of pCCB630 using primers cb886 and cb887 to yield pCCB642 (MBP–SPO11-YFYF). Other SPO11 active site mutants were generated by inverse PCR and self-ligation using template pCCB630. Primers and the resulting plasmids are as follows: SPO11-Y137F (primers cb1577 and cb1578, plasmid pCCB1084), SPO11-Y138F (primers cb1579 and cb1580, plasmid pCCB1085) and SPO11-E224A (primers cb1581 and cb1582, plasmid pCCB1086).

### Expression and purification of recombinant proteins

Viruses were produced using the Bac-to-Bac Baculovirus Expression System (Invitrogen) according to the manufacturer's instructions. We infected $4 \times 10^9$ *Spodoptera frugiperda* Sf9 cells (Gibco, Thermo Fisher) with viruses at a multiplicity of infection of 2. Expression of MBP–SPO11 used viruses generated from pCCB630, and MBP and His–Flag tagged SPO11–TOP6BL complexes used viruses generated from pCCB630 and pCCB628. Following 72 h of infection, cells were collected, washed with PBS, frozen in dry ice and kept at −80 °C until use. All purification steps were carried out at 0–4 °C. Cell pellets were resuspended in 80 ml of lysis buffer (50 mM HEPES-NaOH pH 6.8, 1 mM DTT, 2 mM EDTA and protease inhibitor cocktail (Sigma-Aldrich, catalogue no. P8340) diluted 1:800, supplemented with 4 µM leupeptin, 5.8 µM pepstatin, 6.6 µM chymostatin and 1 mM phenylmethanesulfonyl fluoride), pooled in a beaker and then mixed slowly with a stir bar for 20 min. Ice-cold glycerol (10%) and 1 M NaCl were added to the cell lysate, which was then centrifuged at 43,000*g* for 25 min. The cleared extract was loaded onto 2 ml of pre-equilibrated amylose resin (NEB). The column was washed extensively with amylose buffer (25 mM HEPES-NaOH pH 6.8, 1 M NaCl, 5% glycerol, 1 mM DTT and 2 mM EDTA) and eluted with buffer containing 10 mM maltose. Fractions containing protein were loaded on a HiLoad 16/600 Superdex 200 pg column pre-equilibrated with buffer containing 25 mM HEPES 6.8, 100 mM NaCl, 2 mM DTT and 5 mM EDTA. The peak was collected and diluted twofold in buffer without salt, loaded on a Capto HiRes cation exchange column and eluted with a 0.1–0.5 M NaCl gradient. Fractions containing purified proteins were pooled, and aliquots flash-frozen in liquid nitrogen and stored at −80 °C.

For the MBP and His–Flag tagged SPO11–TOP6BL complexes, Sf9 cells were lysed by raising salt concentration to 500 mM. Cleared extract was incubated for 20 min with 2 ml of pre-equilibrated Ni-NTA resin (Thermo Scientific) and washed extensively with a buffer containing 25 mM HEPES pH 7.5, 500 mM NaCl, 10% glycerol, 0.1 mM DTT and 40 mM imidazole. The complex was eluted with buffer containing 500 mM imidazole and loaded onto 2 ml of equilibrated amylose resin. The resin was washed with 25 mM HEPES 7.5, 500 mM NaCl, 10% glycerol, 1 mM DTT and 2 mM EDTA, and the protein eluted with buffer containing 10 mM maltose. Fractions containing protein were loaded on a Superdex 200 column equilibrated in 25 mM HEPES pH 7.5, 400 mM NaCl, 10% glycerol, 2 mM DTT and 5 mM EDTA. The peak was collected, concentrated using 10-kDa Amicon centrifugal filters (Millipore), aliquoted, flash-frozen in liquid nitrogen and stored at −80 °C.

### SEC–MALS

Light-scattering data were collected using a Superdex 200 increase 10/300 GL SEC column connected to the AKTA Pure Chromatography System (Cytiva). The elution from SEC was monitored using a differential refractometer (Optilab, Wyatt) and a static, dynamic, multiangle laser light-scattering detector (miniDAWN, Wyatt). The SEC–ultraviolet/ light-scattering/refractive index system was equilibrated in buffer containing 25 mM HEPES-NaOH pH 7.5, 500 mM NaCl, 10% glycerol, 5 mM EDTA and 2 mM DTT, at a flow rate of 0.3 ml min⁻¹. Average molecular mass was determined across the entire elution profile at intervals of 0.5 s from static light-scattering measurement using ASTRA software (Wyatt).

### Plasmid cleavage assay

Cleavage reactions (20 µl) were typically carried out with 250 nM MBP–SPO11 and 5 ng µl⁻¹ pUC19-derived 3-kb plasmid (pCCB959) in buffer containing 25 mM Tris pH 7.5, 5% glycerol, 50 mM NaCl, 1 mM DTT, 0.1 mg ml⁻¹ bovine sreum albumin (BSA), 5 mM $MgCl_2$ and 1.5 mM $MnCl_2$, unless stated otherwise. Reactions were incubated for 2 h at 37 °C, stopped with 50 mM EDTA and 1% SDS and treated with 0.2 mg ml⁻¹ proteinase K for 15–30 min at 55 °C. DNA was separated on a 1% TBE-agarose gel and stained using SYBR Gold.

For phenol–chloroform partitioning of cleavage products, cleavage reactions were stopped in the presence or absence of proteinase K. Following 20-min incubation at 55 °C, samples were mixed with an equal volume of phenol–chloroform–isoamyl alcohol and centrifuged for 5 min at 13,000 rpm. The organic phase and interphase were back-extracted twice with 100 mM Tris-HCl pH 8.0, 1 mM EDTA and 200 mM NaCl; the organic and aqueous phases were ethanol precipitated. DNA was resuspended in buffer containing 30 mM Tris-HCl pH 8.5, 1 mM EDTA, 100 mM NaCl and 0.2 mg ml⁻¹ proteinase K, with incubation for 1 h at 55 °C. DNA was again ethanol precipitated, resuspended in TE buffer, separated on 1% TBE-agarose gel and stained using SYBR Gold. Data were quantified using ImageJ2 and plotted with Graph-Pad Prism 9.

For the analysis of DNA cleavage with linear substrates containing zero, one, three or six copies of the Widom 601 sequence, fragments containing one, three or six copies of Widom 601 were cloned into the multiple cloning site of pUC19 to yield pCCB1106, pCCB1107 and pCCB1108, respectively. The plasmids were PCR amplified with primers pl68 and vg001 (containing a 5′ 6-carboxyfluorescein dye) to yield linear substrates for the reaction. Cleavage reactions were performed under standard conditions with 500 nM MBP–SPO11 and 25 ng µl⁻¹ linear fluorescent substrate. DNA was separated on 1% TBE-agarose gel and visualized using a Typhoon scanner (Cytiva).

Nicked DNA substrates were prepared by treatment of pOC157 with Nb.BrsDI, followed by phenol extraction and ethanol precipitation.

### Detection of fluorescent SPO11–DNA covalent complexes

Substrates were assembled by annealing primers dd77 and cb100, or cb1593 and cb100, to produce 80-bp duplex DNA with a 6-FAM fluorophore located in 5′ or 3′, respectively. Oligos were mixed in equimolar concentrations (10 µM) in 100 mM NaCl, 10 mM Tris-HCl pH 8.0 and 1 mM EDTA, heated and slowly cooled on a PCR thermocycler (3 min at 98 °C, 1 h at 75 °C, 1 h at 65 °C, 30 min at 37 °C and 10 min at 25 °C).

Cleavage reactions (20 µl) contained 1 µM MBP–SPO11 and 0.5 µM fluorescent substrate in buffer, containing 25 mM Tris pH 7.5, 5% glycerol, 10% DMSO, 40 mM NaCl, 1 mM DTT, 5 mM $MgCl_2$ and 1.5 mM $MnCl_2$. Reactions were incubated for 2 h at 37 °C, stopped with 1× Leammli buffer and separated by SDS–PAGE. Fluorescent gel was scanned using a Typhoon scanner (Cytiva), and proteins stained with Coomassie blue.

### Sequencing gel analysis of SPO11 cleavage sites

Oligonucleotides cb95 and cb100 were first purified on 10% polyacrylamide-urea gels. For each oligo, 5 pmol was 5′-end labelled with [γ-³²P]ATP and T4 polynucleotide kinase (NEB). The labelled oligo was mixed in equimolar concentrations with the unlabelled reverse complement and annealed by heating at 100 °C in a water bath, followed by slow cooling. Labelled substrates were then purified by native PAGE.

Cleavage reactions (20 µl) contained 500 nM MBP–SPO11, 1 nM radioactive substrate and 2.5 nM (100 ng) plasmid DNA in buffer containing 25 mM Tris pH 7.5, 5% glycerol, 0.1 mg ml$^{-1}$ BSA, 50 mM NaCl, 1 mM DTT, 5 mM MgCl$_2$ and 1.5 mM MnCl$_2$. Reactions were incubated for 2 h at 37 °C then stopped with 50 mM EDTA and 1% SDS. Markers were generated by partial digestion of substrate using either the indicated restriction enzymes or DNase I. DNA was ethanol precipitated and separated on 10% TBE-UREA sequencing gel; the gel was then dried and developed by autoradiography.

### Gel shift assays

The hairpin substrate was assembled by self-annealing of primer cb957. The substrate was 5′ end labelled with [γ-$^{32}$P]ATP (Revvity) and T4 polynucleotide kinase (NEB) and purified by native PAGE. Binding reactions (20 µl) were carried out in 25 mM Tris-HCl pH 7.5, 7.5% glycerol, 50 mM NaCl, 2 mM DTT, 5 mM EDTA, 1 mg ml$^{-1}$ BSA and the indicated amounts of protein complexes. Gel shift assays with radioactive substrates contained 0.1 nM DNA. Reactions were incubated for 30 min at 37 °C and separated on 7% TAE-polyacrylamide/bis (37.5:1) gel at 200 V for 2 h. Gels were dried, exposed to autoradiography plates and demonstrated by phosphorimaging. Gel shift assays with plasmid substrates contained 100 ng DNA. Reactions were incubated for 30 min at 37 °C and separated on 1% TAE-agarose gels at 60 V for 2 h. Gels were stained with SYBR Gold and scanned using a Typhoon scanner (Cytiva).

### Statistics and reproducibility

Sample numbers in quantifications are indicated in the figure legends. Gels shown in the article are representative images. In Fig. 1b,d, ion exchange fractions of purified proteins were analysed by gel electrophoresis and assayed for DNA-cleavage activity at least twice. In Fig. 1e,f, the importance of active site tyrosines and metal ions was confirmed more than three times. In Fig. 2b–e, experiments were performed at least twice, with similar results. In Fig. 3a–c, cleavage sites were mapped on the various substrates at least twice, with similar results. In Fig. 4b,c,e, experiments were reproduced at least twice. In Fig. 1e, DNA-binding activities of the complexes were compared twice, with similar results. In Extended Data Fig. 1a,b, experiments were performed once but the conclusions were confirmed multiple times independently. In Extended Data Fig. 2a, the experiment was performed once. In Extended Data Fig. 3a, the quantification shows a single experiment but the observation was reproduced at least twice under different conditions. In Extended Data Fig. 4d, the observation

was reproduced more than three times. In Extended Data Fig. 5a,b, the observations were reproduced twice under different conditions. In Extended Data Fig. 6a,b, quantifications show a single experiment but the observation was reproduced at least twice under different conditions. In Extended Data Fig. 6f,g, observations were reproduced at least twice under different conditions. In Extended Data Fig. 7a–c, observations were reproduced at least twice. No statistical methods were used to predetermine sample size. Investigators were not blinded to allocation during experiments and outcome assessment.

### Reporting summary

Further information on research design is available in the Nature Portfolio Reporting Summary linked to this article.

## Data availability

AlphaFold 3 models are provided in .pdb format in the Supplementary Information. Gel source data for Figs. 1–5 and Extended Data Figs. 1–7 are provided in Supplementary Fig. 1.

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

**Acknowledgements** We thank S. Keeney for discussion and sharing unpublished information, and B. Hallet and R. Chalmers for comments on the manuscript. We thank C.C.B. laboratory members P. Liloku, D. Daccache, M. Survi and T. C. Borremans for help with SEC–MALS experiments, insect cell maintenance and protein purifications. This work was supported by the European Research Council under the European Union's Horizon 2020 research and innovation programme (ERC grant agreement no. 802525 to C.C.B.). C.O. was funded in part by a fellowship from Fonds National de la Recherche Scientifique (FNRS). C.C.B. is a FNRS Research Associate.

**Author contributions** C.C.B. and C.O. designed the study and secured funding. C.O. purified all proteins and performed experiments in Figs. 1b,d,e, 2c, 4c and 5c and Extended Data Figs. 1, 3 and 7. C.C.B. performed AlphaFold modelling and experiments in Figs. 1f, 2b,d,e, 3, 4b,d,e and 5d,e and Extended Data Figs. 2, 4, 5 and 6. C.C.B. prepared the figures and wrote the paper, with input from C.O.

**Competing interests** The authors declare no competing interests.

**Additional information**
**Correspondence and requests for materials** should be addressed to Corentin Claeys Bouuaert.

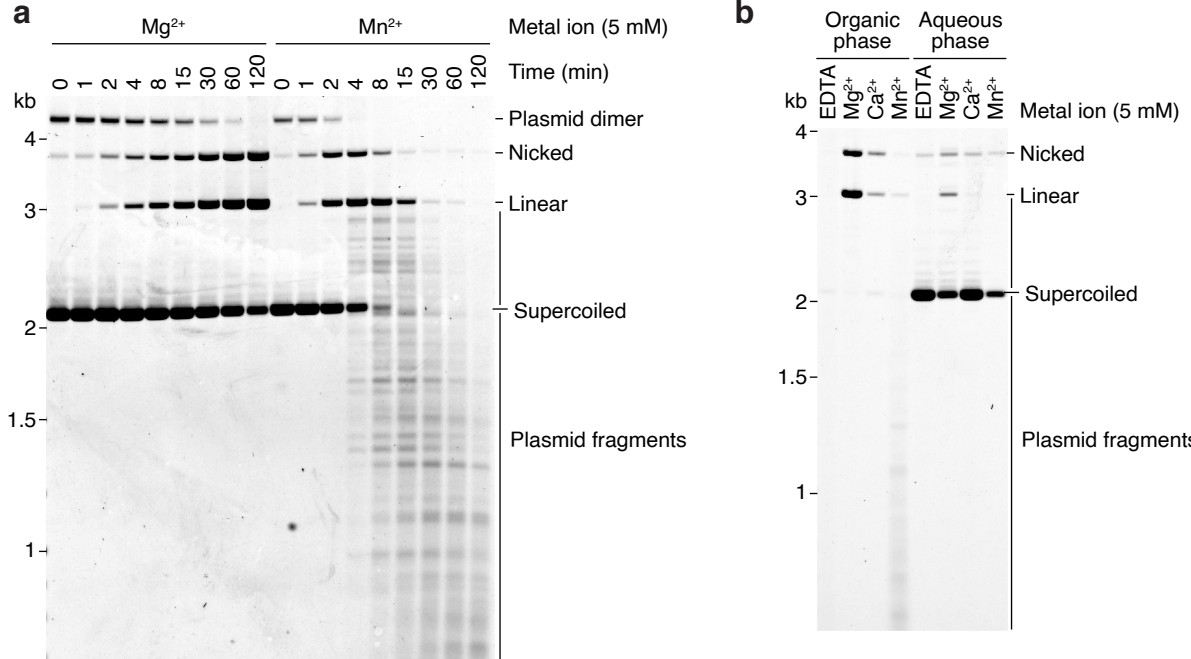

**Extended Data Fig. 1 | Impact of metal ions on SPO11 activity. a**, Kinetic analysis of SPO11 DNA cleavage in the presence of 5 mM $Mg^{2+}$ or $Mn^{2+}$. **b**, Phenol extraction of SPO11 cleavage reactions with EDTA, $Mg^{2+}$, $Ca^{2+}$ or $Mn^{2+}$. Covalent complexes depend on the presence of metal ions, and both single-strand and double-strand cleavage products partition to the organic phase. For gel source data, see Supplementary Fig. 1.

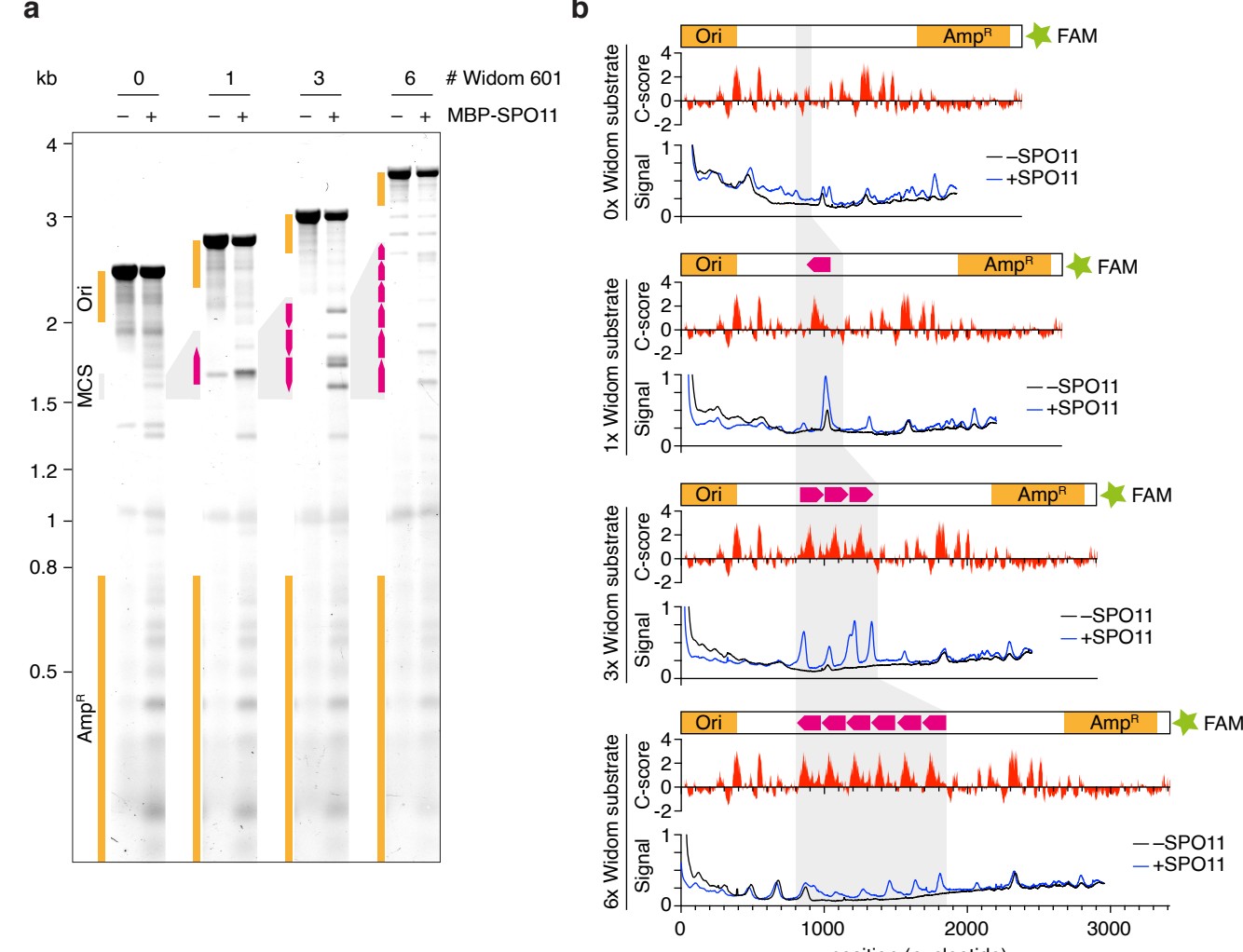

**Extended Data Fig. 2 | Correlation between DNA bendability and SPO11 activity. a**, Agarose gel analysis of SPO11 DNA cleavage with linear substrates containing 0, 1, 3, or 6 copies of the Widom 601 sequence. The pUC19-derived substrates contain a single fluorophore (FAM) at one end, allowing the mapping of cleavage products. The position of the origin of replication (Ori), ampicillin resistance gene (Amp^R) and the multiple cloning site (MCS) are indicated. Copies of the Widom 601 sequence are represented with pink arrows. **b**, Correlation between Widom 601 sequences, DNA bending, and SPO11 cleavage. The C-score is a bendability parameter predicted by DNAcycP[21]. The insertion of Widom 601 sequences creates hotspots for SPO11, although the predicted bendability of the DNA sequence is not sufficient to account for the cleavage activity observed along the substrate. The cleavage hotspots produced by the Widom 601 sequence can also be due to a sequence preference of SPO11, irrespective of bendability. For gel source data, see Supplementary Fig. 1.

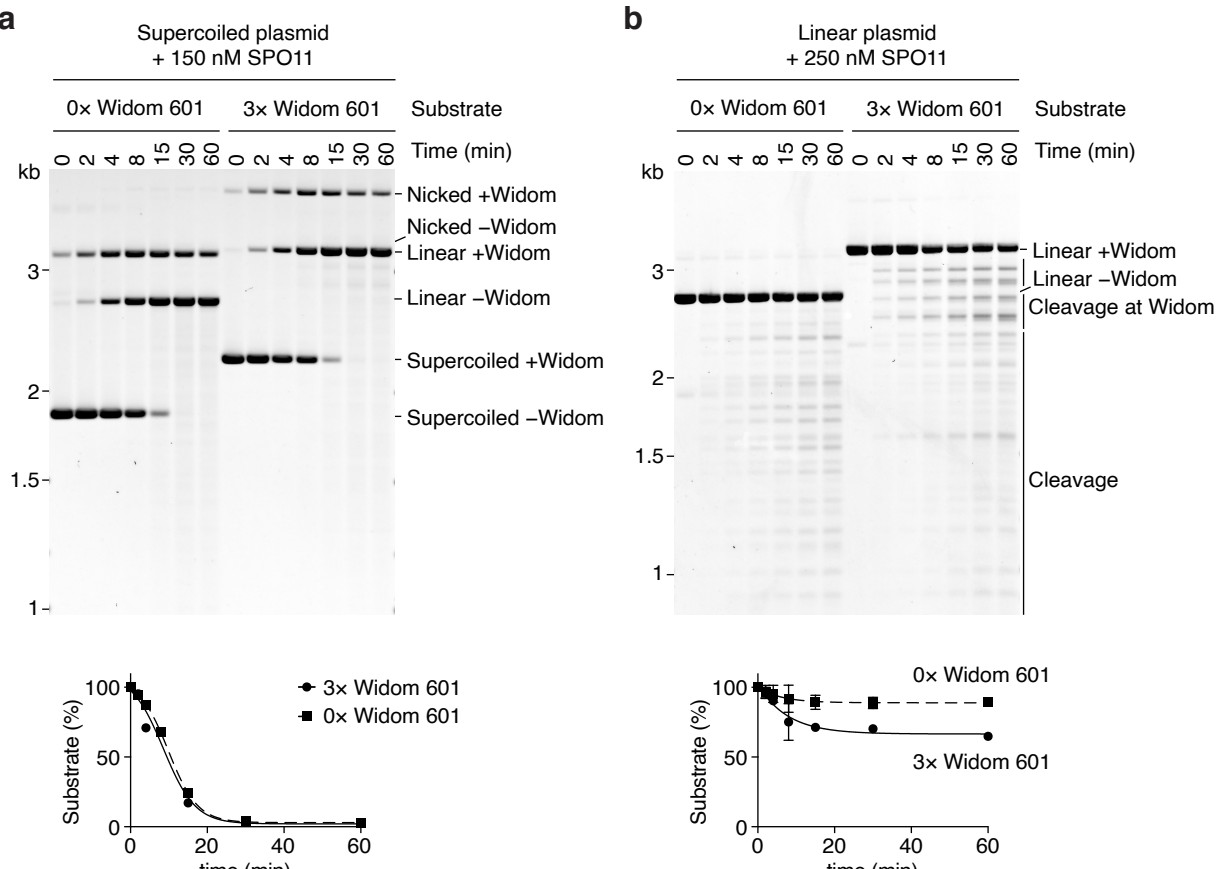

**Extended Data Fig. 3 | Relationship between DNA supercoiling, bending, and cleavage.** Time course analyses of SPO11 cleavage with **(a)** supercoiled and **(b)** linear plasmid DNA substrates without (pUC19) or with three copies of the Widom 601 sequence (pCCB1107). Quantification of the DNA substrate is shown under the gel. On the right, quantifications show the mean and range from two replicates. The two supercoiled substrates are consumed at the the same rate. In contrast, the linear substrate with Widom sequences is cleaved faster than the one without Widom sequences. This suggests that supercoiling accelerates cleavage by facilitating DNA bending. For gel source data, see Supplementary Fig. 1.

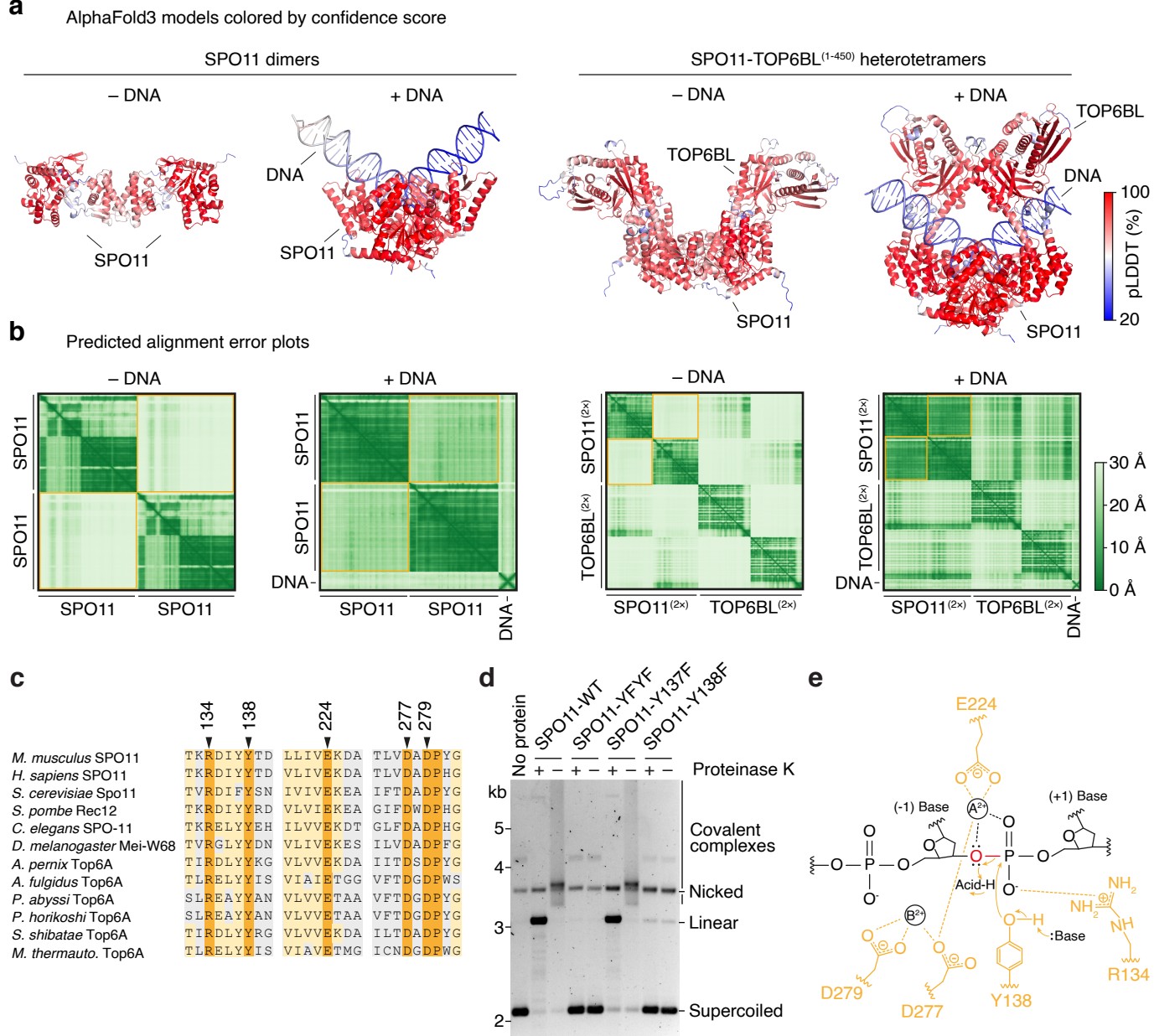

**a** AlphaFold3 models colored by confidence score

**b** Predicted alignment error plots

**c** 

**d** Proteinase K

**e** 

**Extended Data Fig. 4 | Structural modeling of SPO11 complexes and cleavage mechanism. a**, AlphaFold3 models colored by confidence score of SPO11 dimers and SPO11-TOP6BL complexes with or without 40 bp duplex DNA substrate. The C-terminus of TOP6BL was omitted from the model because it is predicted to be unstructured. **b**, Predicted alignment error plots. The structure and relative position (orange squares) of SPO11 monomers are predicted with lower confidence in the absence of DNA than in the presence of DNA, consistent with the monomeric stoichiometry of SPO11 and SPO11-TOP6BL complexes. In the absence of DNA, AlphaFold proposes an aberrant dimeric model of SPO11 throught interactions between WH domains (left). In the presence of DNA and TOP6BL, the relative position of SPO11 is predicted with much higher accuracy (compare orange squares on the rightmost PAE plot with the others).

**c**, Sequence alignments of eukaryotic SPO11 and archaeal Top6A proteins. Blocks around active site residues are shown. Invariant amino acids are in orange, conservative substitutions are in light orange. Active site residues are indicated with an arrowhead. **d**, Cleavage assay with wild-type SPO11, double Y137F/Y138F (YFYF) and single Y137F and Y138F mutants. Conversion of the cleaved linear product into a smear in the absence of proteinase K indicates covalent attachment of SPO11 to the broken DNA ends. The low amount of linear and nicked products observed with the Y138F mutant is due to a non-specific nuclease contaminant in the protein preparation. **e**, Two-metal-ion reaction scheme, based on the mechanism proposed for type IIA topoisomerases[27]. For gel source data, see Supplementary Fig. 1.

**a**

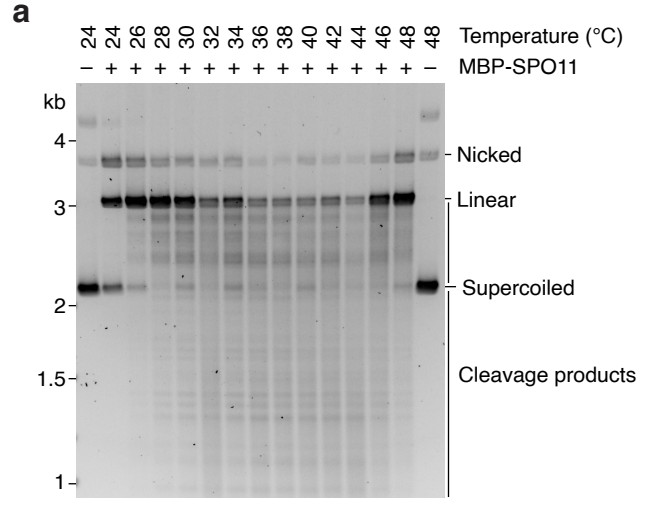

**b**

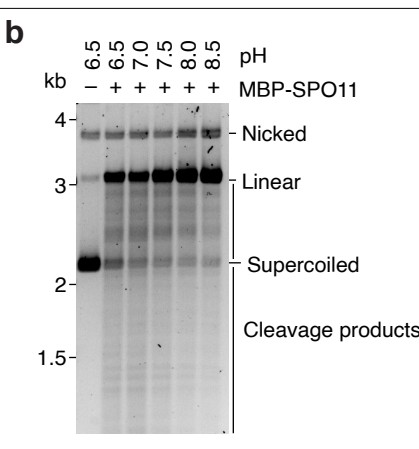

**c**

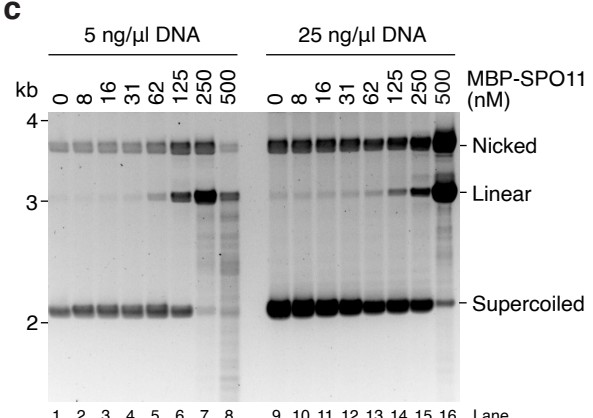

**d**

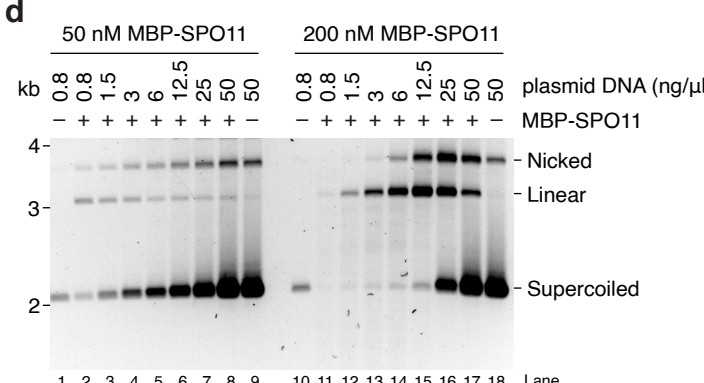

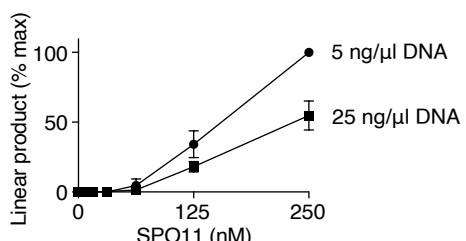

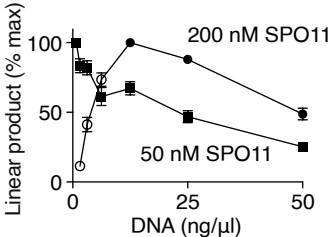

**Extended Data Fig. 5 | Optimization of the cleavage reaction. a**, Effect of the reaction temperature on SPO11 cleavage. **b**, Effect of the pH on SPO11 cleavage. The standard conditions chosen for all the reactions are 37 °C and pH 7.5. **c**, Titration of SPO11 protein in reactions that contained either 5 ng/μl (2.5 nM) or 25 ng/μl (12.5 nM) plasmid DNA. Quantifications show the mean and range from two independent experiments. Cleavage increased with protein concentration, but the total level of cleavage was higher in reactions that had lower DNA concentrations (compare linear product in lanes 5–7 with lanes 13–15). **d**, Titration of DNA in reactions that contained either 50 or 200 nM SPO11. Quantifications show the mean and range from two independent experiments.

At 50 nM SPO11, cleavage was highest at the lowest DNA concentration tested (0.8 ng/μl, 0.4 nM). At 200 nM SPO11, a reduction of total DNA cleavage was observed at substrate concentrations above 12.5 ng/μl (6.2 nM). At lower concentrations (open circles), the substrate is limiting so the amount of linear product is not representative of total break levels. The reaction time in these experiments is 2 h. Note that in Fig. 4d, the inhibitory effect is observed at lower DNA concentrations because the reaction time was shorter (15 min). Longer reaction times provide more opportunity for dimerization and cleavage. For gel source data, see Supplementary Fig. 1.

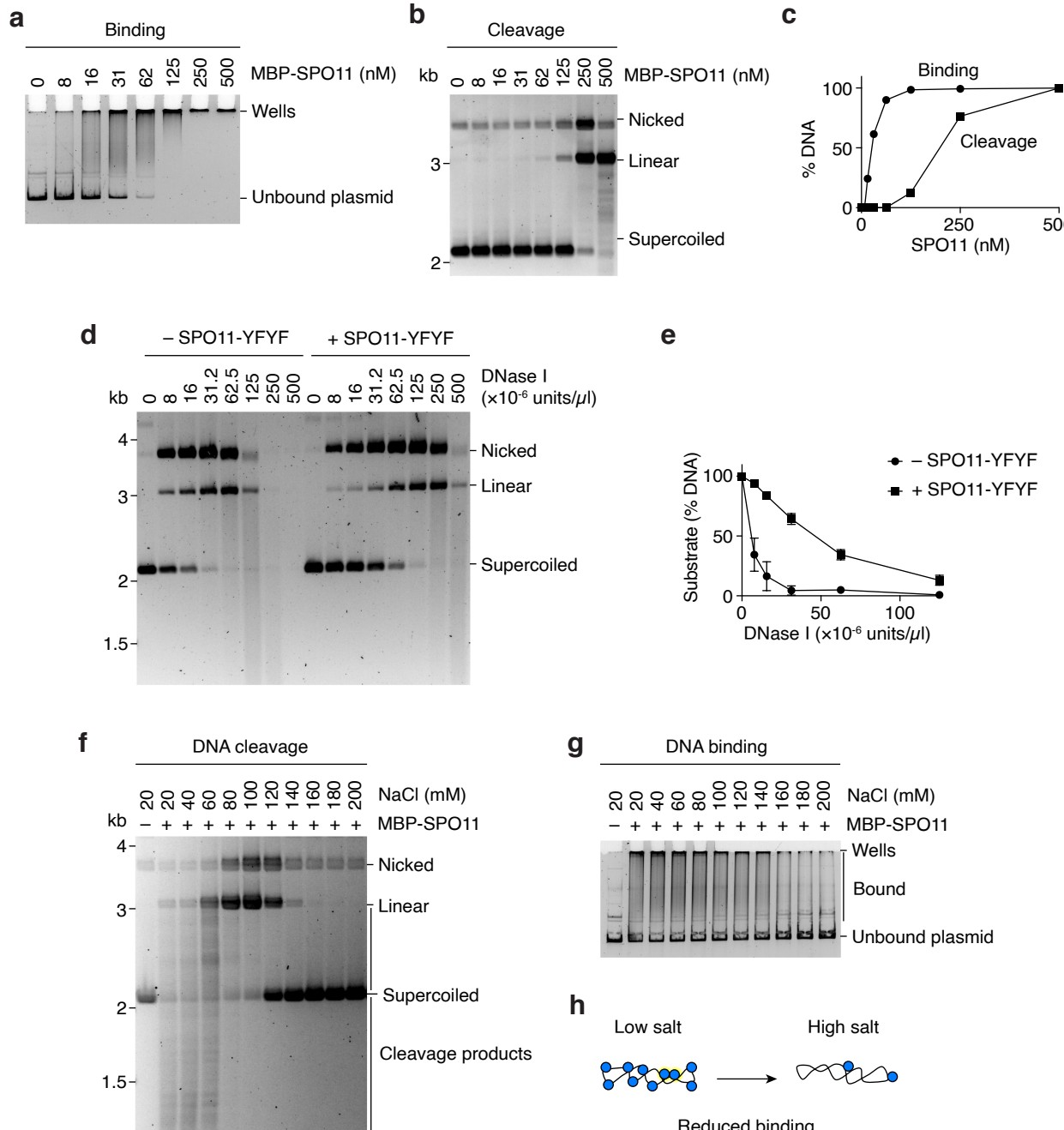

**Extended Data Fig. 6 | Relationships between DNA binding and cleavage.**
**a**,**b**,**c**, DNA cleavage requires higher SPO11 concentration than DNA binding. Comparison of the DNA-binding (a) and DNA cleavage (b) activities of SPO11. Reactions contained 5 ng/µl (2.5 nM) plasmid. Binding reactions were assembled for 30 min. Cleavage reactions were stopped after 2 h. (c) Quantification shows that the plasmid is bound efficiently at concentrations that do not support cleavage. **d**,**e**, SPO11 provides effective protection against DNase I treatment. Plasmid substrates (2.5 nM) were incubated with or without 600 nM catalytically inactive SPO11-Y137F/Y138F mutant, followed by

two-minute treatment with the indicated concentration of DNase I. (e) Quantification of the supercoiled substrate remaining in panel d. Error bars are ranges from two independent experiments. **f**,**g**,**h**, DNA binding and cleavage by SPO11 are sensitive to salt. Effect of NaCl on (f) plasmid cleavage and (g) DNA binding. SPO11 concentration is 500 nM in panel f and 100 nM in panel g. The greater sensitivity of cleavage than DNA binding to salt could be explained by a mild reduction in DNA binding severely reducing the chances of dimerization on DNA (h). For gel source data, see Supplementary Fig. 1.

**a**

Mixture WT & YFYF SPO11
(250 nM total)

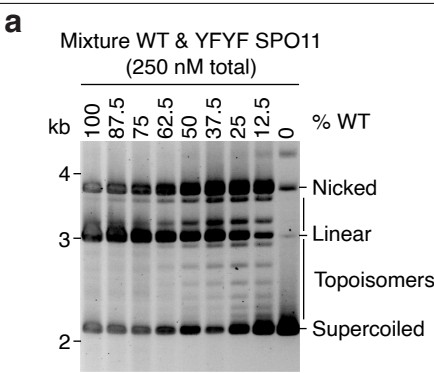

**c**

50 nM SPO11-WT + 250 nM SPO11-YFYF

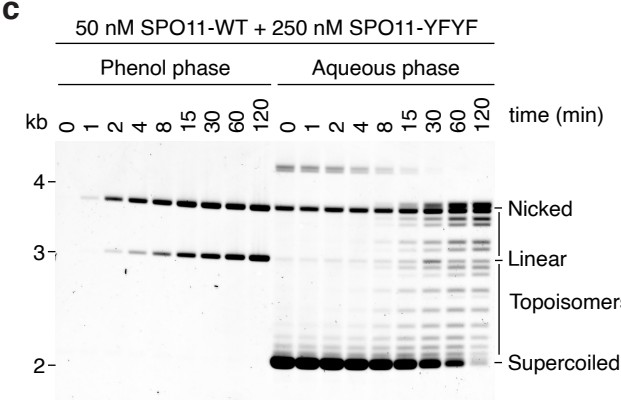

**b**

Equimolar mixture
YFYF & E224A SPO11

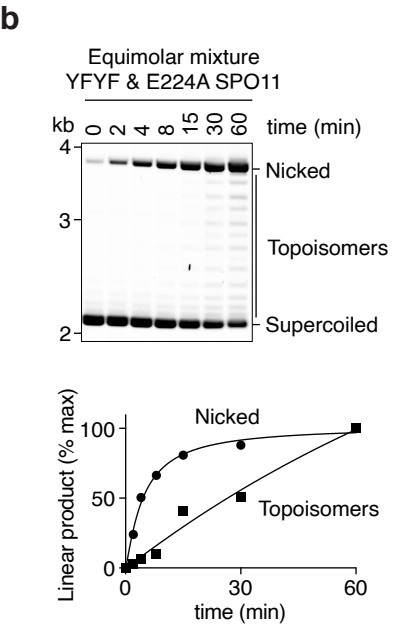

**d**

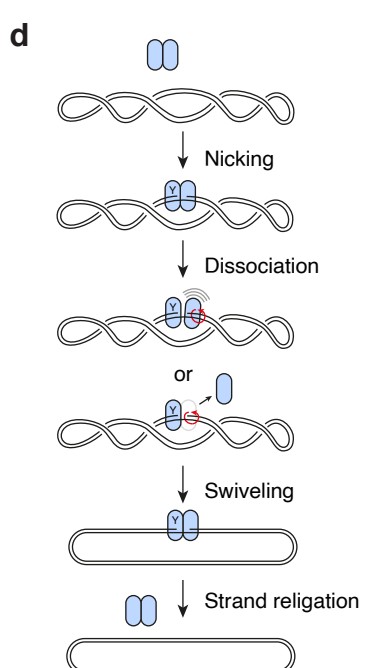

**Extended Data Fig. 7 | SPO11 can reseal single-strand DNA nicks. a**, Cleavage analysis at a constant SPO11 concentration in the presence of different ratios of wild-type and catalytically-inactive (Y137F/Y138F) mutants. The ladder that migrates between the supercoiled and nicked products is absent in reactions that contain only wild-type or inactive mutants, indicating that it is not due to a contaminating activity in one of the protein preparations. **b**, Kinetic analysis of cleavage products in reactions that contained mixtures of YFYF and E224A mutants. **c**, Phenol-chloroform partitioning of DNA cleavage products in reactions that contained mixtures of wild-type and mutant SPO11. Topoisomers partition to the organic phase as the covalent link with SPO11 has been released.

**d**, Illustration of the plasmid relaxation activity observed with mixtures of wild type and inactive SPO11. Dissociation of the SPO11 dimer after single-strand nicking will lead to the swiveling of the DNA duplex around the intact phosphodiester bond (red arrows). Restauration of the SPO11 dimer then provides an opportunity for strand religation. Separation of the dimer interface could also be accompanied by the dissociation of the subunit not involved in catalysis from the DNA substrate, although this could would be expected to lead to full plasmid relaxation, which is not always observed. For gel source data, see Supplementary Fig. 1.

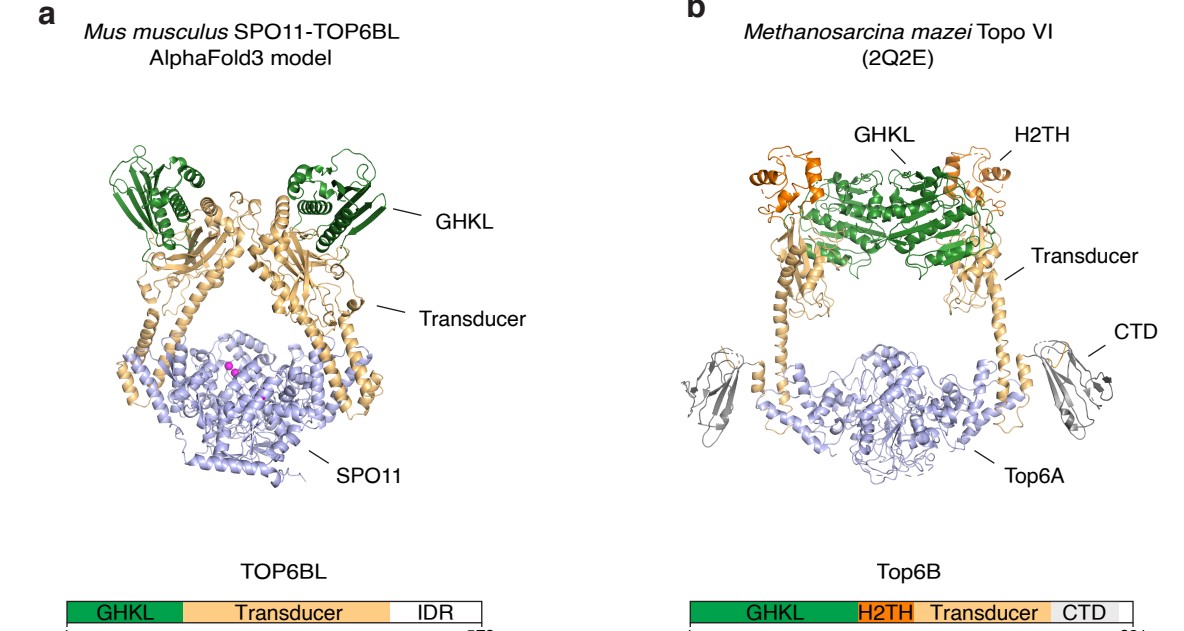

**a**
*Mus musculus* SPO11-TOP6BL
AlphaFold3 model

GHKL

Transducer

SPO11

TOP6BL

| GHKL | Transducer | IDR |

1                                      579

**b**
*Methanosarcina mazei* Topo VI
(2Q2E)

GHKL          H2TH

Transducer

CTD

Top6A

Top6B

| GHKL | H2TH | Transducer | CTD |

1                                      621

**Extended Data Fig. 8 | Comparison of the AlphaFold model of mouse SPO11-TOP6BL and the structure of *Methanosarcina mazei* Topo VI. a**, AlphaFold3 model of a 2:2 SPO11-TOP6BL heterotetramer. The structure was modeled with DNA, but the DNA was hidden to ease comparison. **b**, Crystal structure of Topo VI[11].

SPO11 and Top6A are in blue. GHKL (green), helix two-turn helix (H2TH, orange), transducer (yellow), C-terminal domain (CTD, grey), intrinsically-disordered region (IDR, not shown).

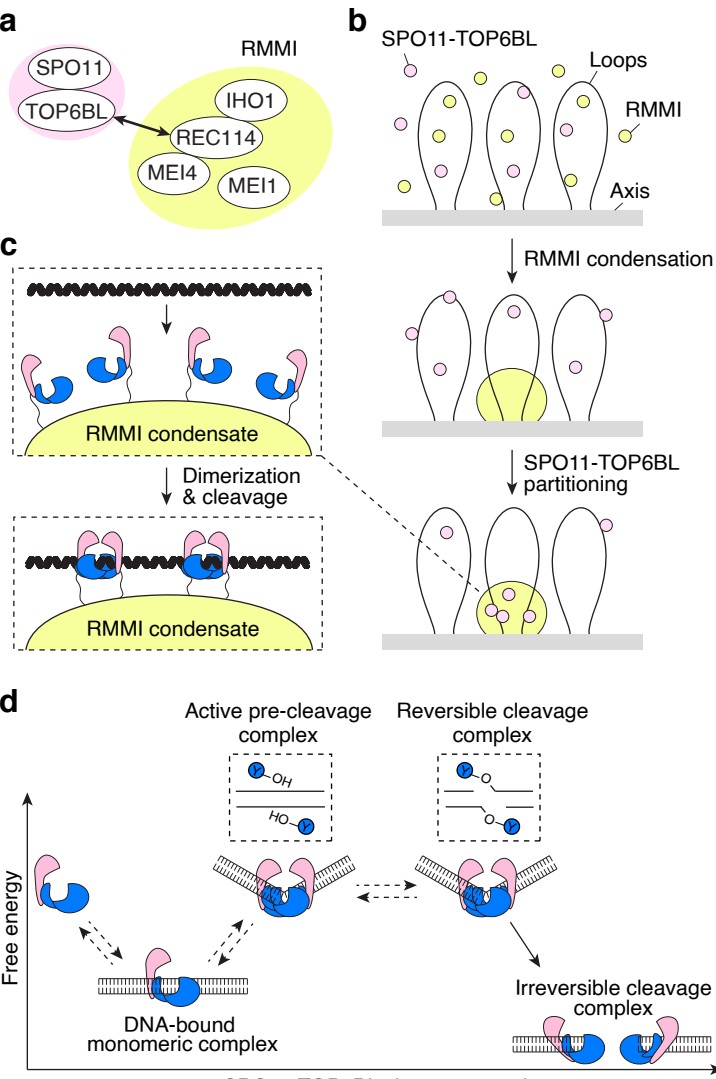

**Extended Data Fig. 9 | Model of meiotic DSB formation in mice. a**, Proteins essential for DSB formation in mice. **b**, We propose that RMMI form condensates along the chromosome axis and recruit SPO11-TOP6BL complexes through an interaction between REC114 and the C-terminus of TOP6BL[31]. The ensuing increase in local concentration of SPO11-TOP6BL complexes allows SPO11 dimerization and cleavage. **c**, Multiple dimers may assemble, leading to the formation of closely-spaced double DSBs[39,40]. **d**, Proposed thermodynamics of SPO11-TOP6BL cleavage reactions. Monomeric SPO11-TOP6BL complexes bind with high affinity to DNA. The assembly of an active pre-cleavage dimeric complex requires bending of the DNA substrate, which would correspond to a high-energy state. Cleavage is isoenergetic and therefore inherently reversible. Conformational transitions caused by or accompanied with the dissociation of SPO11 dimers stabilize SPO11-TOP6BL complexes on DNA ends, creating irreversible breaks.

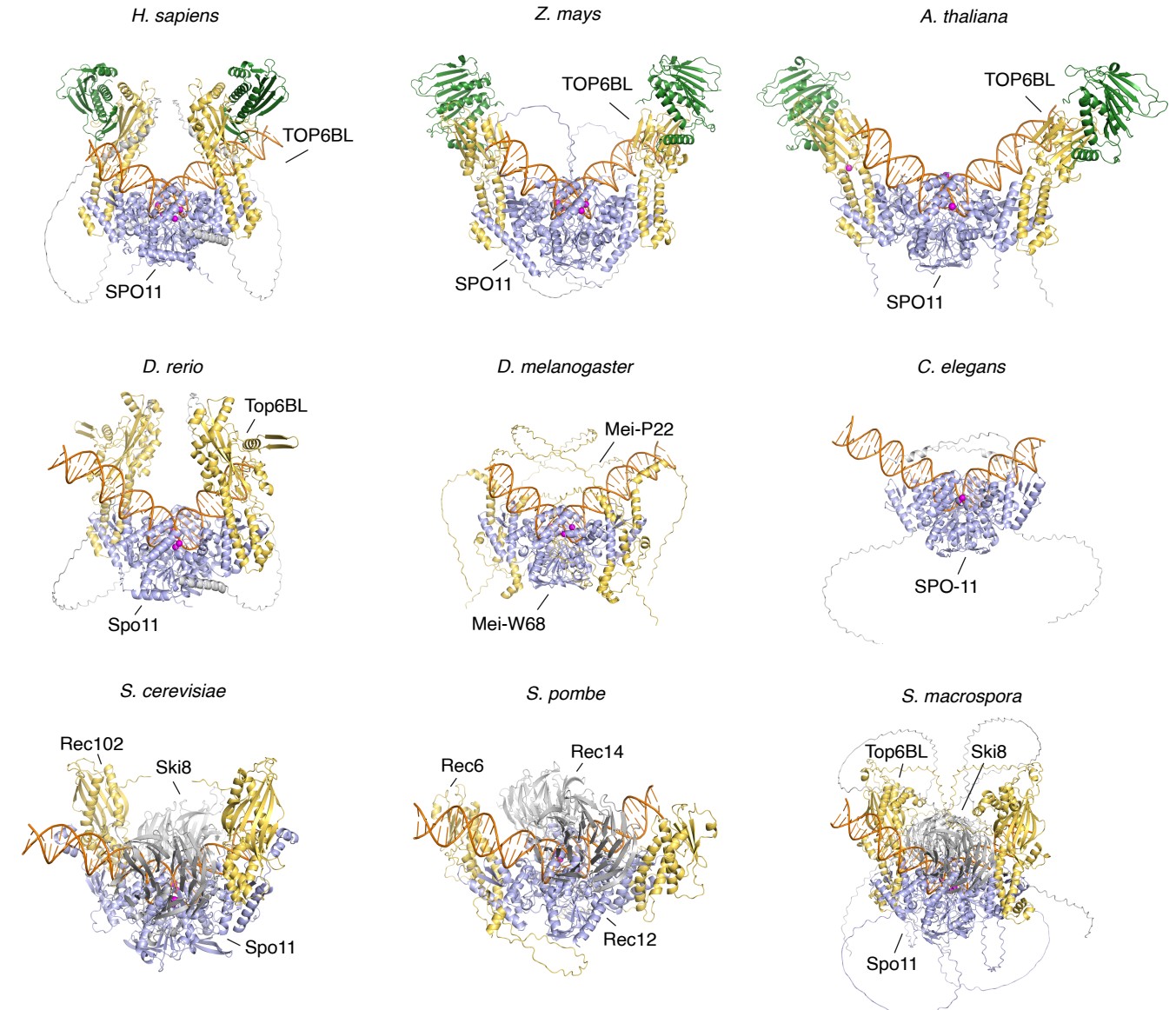

**Extended Data Fig. 10 | AlphaFold models of SPO11 complexes from different species.** AlphaFold3 models of DNA-bound dimeric SPO11 core complexes from *H. sapiens* (Q9Y5K1, Q8N6T0), *Z. mays* (A0A804P805, A0A1D6GVU9), *A. thaliana* (Q9M4A2, Q9M4A1, Q5Q0E6), *D. rerio* (Q6P0S6, B3DIP6), *D. melanogaster* (Q7KPA5, Q9VS36), *C. elegans* (Q22236), *S. cerevisiae* (Spo11, Rec102 and Ski8 from SK1), *S. pombe* (P40384, P40385, Q09150), and *S. macrospora* (Q6WRU4, F7VPQ4 re-annotated[34], Q6URC5). SPO11 homologs are shown in light blue; the transducer domain of TOP6BL homologs are shown in yellow, the GHKL domain in green and C-terminal extension in light grey. Ski8 homologs present in fungal complexes are shown in grey. DNA is shown in orange. Mg$^{2+}$ ions are in magenta. Despite extensive variability in the prediced structure and composition of cleavage complexes, in particular regarding TOP6BL homologs, all the models show SPO11 dimers engaged on a bent duplex DNA substrate, consistent with our interpretation that DNA bending is an important pre-requisite for cleavage.

# Reporting Summary

## Statistics

For all statistical analyses, confirm that the following items are present in the figure legend, table legend, main text, or Methods section.

| n/a | Confirmed | |
|---|---|---|
| ☐ | ☒ | The exact sample size (*n*) for each experimental group/condition, given as a discrete number and unit of measurement |
| ☐ | ☒ | A statement on whether measurements were taken from distinct samples or whether the same sample was measured repeatedly |
| ☒ | ☐ | The statistical test(s) used AND whether they are one- or two-sided<br>*Only common tests should be described solely by name; describe more complex techniques in the Methods section.* |
| ☒ | ☐ | A description of all covariates tested |
| ☒ | ☐ | A description of any assumptions or corrections, such as tests of normality and adjustment for multiple comparisons |
| ☐ | ☒ | A full description of the statistical parameters including central tendency (e.g. means) or other basic estimates (e.g. regression coefficient) AND variation (e.g. standard deviation) or associated estimates of uncertainty (e.g. confidence intervals) |
| ☒ | ☐ | For null hypothesis testing, the test statistic (e.g. *F*, *t*, *r*) with confidence intervals, effect sizes, degrees of freedom and *P* value noted<br>*Give P values as exact values whenever suitable.* |
| ☒ | ☐ | For Bayesian analysis, information on the choice of priors and Markov chain Monte Carlo settings |
| ☒ | ☐ | For hierarchical and complex designs, identification of the appropriate level for tests and full reporting of outcomes |
| ☒ | ☐ | Estimates of effect sizes (e.g. Cohen's *d*, Pearson's *r*), indicating how they were calculated |

*Our web collection on statistics for biologists contains articles on many of the points above.*

## Software and code

Policy information about availability of computer code

| | |
|---|---|
| Data collection | SEC-MALS data collection used ASTRA software version 8 (Wyatt Corp., Santa Barbara, CA). |
| Data analysis | Gels were quantified using ImageJ2 and plotted using GraphPad Prism (version 9). Structure predictions were made with AlphaFold3. |

For manuscripts utilizing custom algorithms or software that are central to the research but not yet described in published literature, software must be made available to editors and reviewers. We strongly encourage code deposition in a community repository (e.g. GitHub). See the Nature Portfolio guidelines for submitting code & software for further information.

## Data

Policy information about availability of data

All manuscripts must include a data availability statement. This statement should provide the following information, where applicable:
- Accession codes, unique identifiers, or web links for publicly available datasets
- A description of any restrictions on data availability
- For clinical datasets or third party data, please ensure that the statement adheres to our policy

AlphaFold3 models are provided in .pdb format in Supplementary Data. Source data are provided with this paper.

## Research involving human participants, their data, or biological material

Policy information about studies with human participants or human data. See also policy information about sex, gender (identity/presentation), and sexual orientation and race, ethnicity and racism.

| | |
|---|---|
| Reporting on sex and gender | Not applicable |
| Reporting on race, ethnicity, or other socially relevant groupings | Not applicable |
| Population characteristics | Not applicable |
| Recruitment | Not applicable |
| Ethics oversight | Not applicable |

Note that full information on the approval of the study protocol must also be provided in the manuscript.

## Field-specific reporting

Please select the one below that is the best fit for your research. If you are not sure, read the appropriate sections before making your selection.

☒ Life sciences ☐ Behavioural & social sciences ☐ Ecological, evolutionary & environmental sciences

For a reference copy of the document with all sections, see nature.com/documents/nr-reporting-summary-flat.pdf

## Life sciences study design

All studies must disclose on these points even when the disclosure is negative.

| | |
|---|---|
| Sample size | No sample size calculations were performed. Sample sizes were chosen based on established best practices in the field for experimental methods used. |
| Data exclusions | No samples were excluded. |
| Replication | All conclusions described in the paper were based on findings reproduced in replicate experiments, except as specified in Statistics and reproducibility section. |
| Randomization | Not relevant. All experiments involved direct comparisons between experimental and control samples. |
| Blinding | Not relevant. All experiments involved direct comparisons between experimental and control samples. |

## Reporting for specific materials, systems and methods

We require information from authors about some types of materials, experimental systems and methods used in many studies. Here, indicate whether each material, system or method listed is relevant to your study. If you are not sure if a list item applies to your research, read the appropriate section before selecting a response.

### Materials & experimental systems

| n/a | Involved in the study |
|---|---|
| ☒ | ☐ Antibodies |
| ☐ | ☒ Eukaryotic cell lines |
| ☒ | ☐ Palaeontology and archaeology |
| ☒ | ☐ Animals and other organisms |
| ☒ | ☐ Clinical data |
| ☒ | ☐ Dual use research of concern |
| ☒ | ☐ Plants |

### Methods

| n/a | Involved in the study |
|---|---|
| ☒ | ☐ ChIP-seq |
| ☒ | ☐ Flow cytometry |
| ☒ | ☐ MRI-based neuroimaging |

# Eukaryotic cell lines

Policy information about cell lines and Sex and Gender in Research

| | |
|---|---|
| Cell line source(s) | Spodoptera frugiperda Sf9 cells for expression of recombinant proteins were from Gibco (Thermo Fisher 11496015). |
| Authentication | Sf9 certificates of analyses are available at https://www.thermofisher.com/order/catalog/product/11496015#/11496015. |
| Mycoplasma contamination | We did not test the cell line for mycoplasma contamination. Sf9 cells were acquired from Gibco. Mycoplasma testing of Sf9 cells is referenced at https://www.thermofisher.com/order/catalog/product/11496015#/11496015 |
| Commonly misidentified lines (See ICLAC register) | No commonly misidentified cell lines were used in the study. |

# Plants

| | |
|---|---|
| Seed stocks | *Report on the source of all seed stocks or other plant material used. If applicable, state the seed stock centre and catalogue number. If plant specimens were collected from the field, describe the collection location, date and sampling procedures.* |
| Novel plant genotypes | *Describe the methods by which all novel plant genotypes were produced. This includes those generated by transgenic approaches, gene editing, chemical/radiation-based mutagenesis and hybridization. For transgenic lines, describe the transformation method, the number of independent lines analyzed and the generation upon which experiments were performed. For gene-edited lines, describe the editor used, the endogenous sequence targeted for editing, the targeting guide RNA sequence (if applicable) and how the editor was applied.* |
| Authentication | *Describe any authentication procedures for each seed stock used or novel genotype generated. Describe any experiments used to assess the effect of a mutation and, where applicable, how potential secondary effects (e.g. second site T-DNA insertions, mosiacism, off-target gene editing) were examined.* |

