## [Peer Review File · Nature]

SPO11 dimers are sufficient to catalyze DNA double-strand breaks in vitro

Corresponding Author: Dr Corentin Claeys Bouuaert

Version 0:

Reviewer comments:

Referee #1

(Remarks to the Author)

The manuscript "SPO11 dimerization controls meiotic DNA double-strand break formation" by Over and Claeys Bouuaert describes the reconstitution of SPO11-mediated DSB formation in vitro. This can fairly be described as a holy grail in the meiosis field, and the authors should be commended for an exceptionally clean, clear, and concisely presented work. Given the importance of the work and the clarity of the experiments and presentation, I have very few comments and questions. After the authors address the comments below, I heartily support publication.

Comments/questions:

Overall, this paper feels written in a very clear, concise manner that would appeal to the half-dozen or so people who have thought about Topo VI and Spo11 for longer than we care to admit (and it does a wonderful job at that!). However: for a general reader, some additional explanations, especially of the more obscure experiments and the reasoning behind them, would be helpful throughout the manuscript. Related to this point, I feel that the addition of some schematics would benefit the broad appeal of this paper, especially for the experiments in Figs 1 and 2.

The results with SPO11-mediated cleavage of the Widom 601 sequence are interesting. I agree that there is likely a correlation between DNA bendability and SPO11 target preference, but the C-score plots and cleavage plots clearly do not line up in Ext. Data Fig. 1. This suggests that the C-score is not capturing the most important determinant of SPO11 site selection, be it bendability or other sequence preference. The authors do mention this on p. 5, but I feel like the phrase "bendability alone as predicted by DNACycP is not sufficient to account for the target site preference of SPO11" does not capture the truth - in fact the C-score seems almost anti-correlated with cleavage propensity within each Widom repeat. Are there other bendability metrics that might correlate better with cleavage preference?

What does the bendability profile look like for the 80-base linear substrate? I played with this myself a bit and a couple of the sites (around bases 30 and 48 do seem to line up with high bendability (if you give the server three repeats of the sequence so it predicts propensity across at least one entire repeat). But the other sites do not align with highly bendable sequences as predicted by this server...

Fig. 4c: Please add some visual indication of which gel fractions come from which regions of the chromatography trace.

I feel like the experiment in Fig. 4e needs more explanation. The finding that only nicks are created in this experiment shows that the newly-added WT SPO11 is (effectively) all dimerizing with pre-bound YYFF monomers. This is to be expected, but it is not very clear that this is what is happening from the explanation of this experiment on p. 6. (By the way, the schematics in Fig. 4 are great and really aid understanding - though I would somehow highlight the heterodimer making the nick in the right-hand bottom schematic in 4e).

FYI, not all Top6B proteins have the "structured C-terminal extension that are absent in TOP6BL." (p. 7); in fact I believe it's a minority of T6B proteins that have this domain. Of the two known Topo VI holoenzyme structures, the enzyme from *M. mazei* (PDB ID 2Q2E) has this domain, but the enzyme from *S. shibatae* (PDB ID 2ZBK) does not.

Fig. 6a: It would help to color SPO11 and TOP6BL consistently with panel b (or alternatively, label SPO11 and TOP6B in

panel b).

Referee #2

(Remarks to the Author)

It has long been acknowledged that the SPO11 protein, related to TopoVI, is the catalytic subunit responsible for DSB formation during meiosis, but for 27 years there had been no demonstration that SPO11 can make DSBs in vitro. With this paper and an accompanying paper from Zheng et al., the long wait is over. Using the mouse SPO11 alone, the authors have detected several activities in addition to the DNA-binding activity already shown for the yeast protein: double-strand cleavage; single-strand nicking; and supercoil relaxation. The in vitro activity has properties expected from previous in vivo studies, mostly in yeast: SPO11 is bound covalently to the cleavage product and cleavage requires two hybrid sites. In addition, the data indicate that bent DNA is a preferred substrate, that nicks as well as DSBs are formed, and that supercoils can be relaxed by the enzyme, consistent with nick resealing--all conclusions that were also made by Zheng et al. The major unique contribution of this manuscript is that the SPO11 dimer alone is cleavage-competent; Zheng et al. focused exclusively on the SPO11-TOB6BL heterotrimer as the active entity.

This is an important advance in the field, and it almost certainly will be cited in every paper on meiotic recombination for the foreseeable future. It opens the door for further biochemical characterization of SPO11's cleavage activity and its regulation. Because the conclusions rely exclusively on behavior of the complex in vitro, and on in silico modeling of the complex's structure, it will be important in the future to provide in vivo validation of the conclusion and to obtain real structures of the proteins and substrate. Now that the door has been opened, it is likely that many will be rushing through.

Comments:

1. Many of the experiments assay reactions at a single, extended time point (2h). It would be useful to include a time-course of the reaction, as is done with the 24x Widom 601 substrate. For example, it appears that the reaction is nearly as efficient in Mg⁺⁺ as it is in Mn⁺⁺, but this is most likely due to the extended incubation time used. A kinetic comparison would seem important here.
2. As another example, a comparison of the nicking and cleavage kinetics for pUC19 versus the 24x Widom 601 substrate might reinforce the importance for bending, as cleavage appears to continue with the Widom 601 substrate long after supercoiled plasmids have disappeared—would this be the case with pUC19?
3. The Mg⁺⁺-cleaved product should be used for the phenol-partition experiment, to prove that the nicked products also have covalently-linked SPO11.
4. All products, not just linear products, should be separately quantified and reported (in supplement if necessary).
5. The main message of the title cannot be said to be proven by the current data. I suggest something that is more indicative of the content, such as "Mouse SPO11 dimers form double-strand breaks in vitro".

Referee #3

(Remarks to the Author)

Oger et al. successfully reconstituted SPO11 complex and characterized the cleavage activity in vitro. They found SPO11 catalyzes DSB formation and then covalently attached to the 5' broken strands in the absence of cofactors. Interestingly, its cleavage site selection is influenced by base composition, DNA bendability, and topology. It seems also that SPO11 can reseal DNA nicks. These results are very interesting and provide valuable information for further understanding the regulation of meiotic DSBs.

It is necessary to have a comprehensive discussion of the cleavage differences (target DNA and SPO11 complex activity) observed from in vitro and in vivo experiments.

It would be better to compare/discuss the conservation and difference for SPO11 complex in different organisms, such as yeast, human, and mouse (maybe with the help of AlphaFold prediction?).

Line numbers are required for review.

Page 4, about middle: "As expected", please clarify the rationale.

Page 5, the first two paragraphs: Why prefer to bending and supercoiled DNA? Is this related to the observations that supercoiling affects DSBs and crossovers enriched in negative supercoiled regions in yeast (PMID: 36107772; PMID: 32152049)?

Page 6, "Dimerization controls DNA cleavage" and "SPO11 can reseal single-strand nicks": The data do not support these conclusions, please consider to revise them.

Page 9, the last paragraph: Given a dimer is required for high cleavage activity, why is there only weak dimer interface?

Fig. 1d,e: What is the top band in the gel? 1f, a doublet for nicked bands?

Fig. 2a: Supercoiled, nicked, and linear all look like doublets?

Version 1:

Reviewer comments:

Referee #1

(Remarks to the Author)

The authors have done a good job addressing all comments. I support publication.

Referee #2

(Remarks to the Author)

The authors' revision has nicely addressed all of my scientific concerns and I fully support publication of this important paper in Nature. I still think that the title is both premature and a bit off topic--the major accomplishment here is the demonstration that Spo11 alone can form DSBs in vitro. But I leave that to the editors and authors to sort out.

Referee #3

(Remarks to the Author)

The authors have resolved my concerns. Congratulations.

Minor comments: The authors should add their interpretation to the doublets in Figures 1 and 2 into the legends.

Oger and Claeys Bouuaert. Response to reviewer's comments.

Reviewer's comments are in back; our responses are in blue.

Referee #1:

The manuscript "SPO11 dimerization controls meiotic DNA double-strand break formation" by Over and Claeys Bouuaert describes the reconstitution of SPO11-mediated DSB formation in vitro. This can fairly be described as a holy grail in the meiosis field, and the authors should be commended for an exceptionally clean, clear, and concisely presented work. Given the importance of the work and the clarity of the experiments and presentation, I have very few comments and questions. After the authors address the comments below, I heartily support publication.

We thank the reviewer for the kind words and encouraging feedback.

Comments/questions:

Overall, this paper feels written in a very clear, concise manner that would appeal to the half-dozen or so people who have thought about Topo VI and Spo11 for longer than we care to admit (and it does a wonderful job at that!). However: for a general reader, some additional explanations, especially of the more obscure experiments and the reasoning behind them, would be helpful throughout the manuscript. Related to this point, I feel that the addition of some schematics would benefit the broad appeal of this paper, especially for the experiments in Figs 1 and 2.

We agree that some experiments deserve more explanation. In the revised manuscript, we sought to follow the reviewers' recommendation, keeping in mind that the short format of the paper forces us to make judgement calls regarding the level of detail that we can add.

We have provided additional explanations to describe the in vitro assay and the cleavage products observed (L70-75), and added cartoons in Figures 1 and 2.

The results with SPO11-mediated cleavage of the Widom 601 sequence are interesting. I agree that there is likely a correlation between DNA bendability and SPO11 target preference, but the C-score plots and cleavage plots clearly do not line up in Ext. Data Fig. 1. This suggests that the C-score is not capturing the most important determinant of SPO11 site selection, be it bendability or other sequence preference. The authors do mention this on p. 5, but I feel like the phrase "bendability alone as predicted by DNACycP is not sufficient to account for the target site preference of SPO11" does not capture the truth - in fact the C-score seems almost anti-correlated with cleavage propensity within each Widom repeat. Are there other bendability metrics that might correlate better with cleavage preference?

Indeed, the preferred cleavage site does not align with the center of the Widom sequence and maps instead to one extremity of the repeats, which does not correlate with the highest C score. We do not know whether this pattern reflects a preference for a specific geometry, sequence, or both.

We have edited the text to clarify this point, which now reads (L136): "Nevertheless, the peak of DNA bendability did not align with preferred cleavage site and some sites predicted to be bendable did not produce strong cleavage sites, indicating that bendability alone as predicted by DNACycP is not sufficient to account for the target site preference of SPO11."

We agree that C-score is not an ideal metric. First, DNA bendability that promotes self-ligation of a DNA segment may not be the same as bendability as sensed by SPO11. Second, we do not envision DNA geometry to be the only factor determining SPO11 cleavage site selection, so we doubt that an ideal metric would exist.

Following the reviewer's question, we have nevertheless searched for other tools. The BendNet server also predicts bendability using an algorithm trained on sequencing of DNA circles like DNAcycP. Not surprisingly, the outputs are very similar. Since the conclusions and limitations are identical, we have not included this in the revised paper.

Instead, we have added a Supplementary Discussion to describe in detail our understanding of the factors that impact target site selection (L697-731).

What does the bendability profile look like for the 80-base linear substrate? I played with this myself a bit and a couple of the sites (around bases 30 and 48 do seem to line up with high bendability (if you give the server three repeats of the sequence so it predicts propensity across at least one entire repeat). But the other sites do not align with highly bendable sequences as predicted by this server...

This is what we find too. However, the C-score obtained using this substrate ranges from -0.4 to 1, which is a much narrower range than obtained for the Widom sequence analysis. More importantly, we think that the experiment with the 80-bp linear substrate is not very adequate to address this question in detail. First because the substrate is linear and is as such inherently a poor substrate. Second because it is short and used at very high protein:DNA ratio, hence we think that the binding dynamics and positioning on the substrate are likely to be impacted by the extremities. Our take is that the selection of cleavage sites is driven by a combination of factors, including sequence, bendability and underwinding. However, other factors come into play in the in vitro assay, notably the stringency of the reaction.

We elaborate our argument of the combined effects of sequence, bendability and underwinding in the Supplementary Discussion (L697-731).

Fig. 4c: Please add some visual indication of which gel fractions come from which regions of the chromatography trace.

Done.

I feel like the experiment in Fig. 4e needs more explanation. The finding that only nicks are created in this experiment shows that the newly-added WT SPO11 is (effectively) all dimerizing with pre-bound YYFF monomers. This is to be expected, but it is not very clear that this is what is happening from the explanation of this experiment on p. 6. (By the way, the schematics in Fig. 4 are great and really aid understanding - though I would somehow highlight the heterodimer making the nick in the right-hand bottom schematic in 4e).

We have added the following sentence to the text (L218): "This mostly generates nicked products, indicating that cleavage is caused by heterodimerization of wild-type SPO11 with pre-bound inactive subunits."

As suggested, we also highlighted the active dimer in Figure 4e and 4d.

FYI, not all Top6B proteins have the "structured C-terminal extension that are absent in TOP6BL." (p. 7); in fact I believe it's a minority of T6B proteins that have this domain. Of the two known Topo VI holoenzyme structures, the enzyme from *M. mazei* (PDB ID 2Q2E) has this domain, but the enzyme from *S. shibatae* (PDB ID 2ZBK) does not.

Thanks for the reminder. This has been corrected (L249).

Fig. 6a: It would help to color SPO11 and TOP6BL consistently with panel b (or alternatively, label SPO11 and TOP6B in panel b).

Done, thank you.

Referee #2:

It has long been acknowledged that the SPO11 protein, related to TopoVI, is the catalytic subunit responsible for DSB formation during meiosis, but for 27 years there had been no demonstration that SPO11 can make DSBs in vitro. With this paper and an accompanying paper from Zheng et al., the long wait is over. Using the mouse SPO11 alone, the authors have detected several activities in addition to the DNA-binding activity already shown for the yeast protein: double-strand cleavage; single-strand nicking; and supercoil relaxation. The in vitro activity has properties expected from previous in vivo studies, mostly in yeast: SPO11 is bound covalently to the cleavage product and cleavage requires two hybrid sites. In addition, the data indicate that bent DNA is a preferred substrate, that nicks as well as DSBs are formed, and that supercoils can be relaxed by the enzyme, consistent with nick resealing--all conclusions that were also made by Zheng et al. The major unique contribution of this manuscript is that the SPO11 dimer alone is cleavage-competent; Zheng et al. focused exclusively on the SPO11-TOP6BL heterotrimer as the active entity.

This is an important advance in the field, and it almost certainly will be cited in every paper on meiotic recombination for the foreseeable future. It opens the door for further biochemical characterization of SPO11's cleavage activity and its regulation. Because the conclusions rely exclusively on behavior of the complex in vitro, and on in silico modeling of the complex's structure, it will be important in the future to provide in vivo validation of the conclusion and to obtain real structures of the proteins and substrate. Now that the door has been opened, it is likely that many will be rushing through.

Comments:

1. Many of the experiments assay reactions at a single, extended time point (2h). It would be useful to include a time-course of the reaction, as is done with the 24x Widom 601 substrate. For example, it appears that the reaction is nearly as efficient in Mg⁺⁺ as it is in Mn⁺⁺, but this is most likely due to the extended incubation time used. A kinetic comparison would seem important here.

This experiment is now presented in **Extended Data Fig. 1a**. As expected, the kinetic analysis indicates that cleavage is much more effective with Mn²⁺ than Mg²⁺.

2. As another example, a comparison of the nicking and cleavage kinetics for pUC19 versus the 24x Widom 601 substrate might reinforce the importance for bending, as cleavage appears to continue with the Widom 601 substrate long after supercoiled plasmids have disappeared—would this be the case with pUC19?

Thanks for the suggestion. The pUC19 and 24x Widom 601 plasmids are too different for a fair comparison, so we opted to compare pUC19 with the derivative with 3 copies of Widom601 (pCCB1107). We performed experiments with supercoiled and linear substrates (**Extended Data Fig. 3**). The results show that the supercoiled substrates with and without Widom 601 sequences were cleaved at the same rate. In contrast, with the linear substrate, the presence of Widom sequences accelerated cleavage. The interpretation is that supercoiled substrates are cleaved at the same rate because supercoiling allows efficient DNA bending and catalysis. However, in the absence of supercoiling, the presence of intrinsically bendable sequences improves cleavage. This reinforces our conclusion that DNA bending is important for cleavage.

3. The Mg⁺⁺-cleaved product should be used for the phenol-partition experiment, to prove that the nicked products also have covalently-linked SPO11.

This experiment is now presented in Extended Data Fig 1b. Phenol extraction confirms that products of single and double-strand cleavage have covalently-bound Spo11, regardless of the metal ions used.

4. All products, not just linear products, should be separately quantified and reported (in supplement if necessary).

We would be happy to follow the reviewers' suggestion, but it is not clear what experiments the reviewer is referring to and what purpose the quantification serves. Any plots added to the paper should convey a message, so we need to understand what information we are trying to extract before plotting data. In the case of the kinetic analysis in Fig 3d for example, we chose not to plot the supercoiled and nicked products, because they are not comparable between the two sides of the gel and plotting them is more likely to mislead the reader than to help. We chose to focus on the linear product rather than the sum of all cleavage products to minimize quantification error, and reported quantifications of early time points to minimize the error caused by secondary cleavage events.

The linear range in these experiments is rather poor. It is better than a Western blot but much worse than phosphorimaging. If one were to quantify all the products in Fig 3d, the total amount of signal would be higher at late time points than at time zero. This is a consequence of (i) unequal binding of intercalating agent to supercoiled and relaxed DNA, and (ii) mediocre signal-to-noise ratio. For these reasons, we are not in favor of quantifying DNA products blindly, but we chose to quantify specific products as appropriate to best convey the key result of the experiment.

5. The main message of the title cannot be said to be proven by the current data. I suggest something that is more indicative of the content, such as "Mouse SPO11 dimers form double-strand breaks *in vitro*".

We understand the point raised by the reviewer: the manuscript does not demonstrate that SPO11 dimerization limits cleavage *in vivo*. That is right. Having said that, to what extent the biochemistry informs biology is a matter of perspective. As biochemists, we would argue it very much does.

We are not convinced by the title proposed by the reviewer as it fails to convey the central message of the paper. Our second choice was something along the lines of: 'In vitro reconstitution of DNA double-strand break formation by SPO11', but we opted against it because the title is too similar to the one chosen by the Keeney lab. We propose to leave it up to the editors to decide whether they object to our current title.

Referee #3:

Oger et al. successfully reconstituted SPO11 complex and characterized the cleavage activity *in vitro*. They found SPO11 catalyzes DSB formation and then covalently attached to the 5' broken strands in the absence of cofactors. Interestingly, its cleavage site selection is influenced by base composition, DNA bendability, and topology. It seems also that SPO11 can reseal DNA nicks. These results are very interesting and provide valuable information for further understanding the regulation of meiotic DSBs.

It is necessary to have a comprehensive discussion of the cleavage differences (target DNA

and SPO11 complex activity) observed from in vitro and in vivo experiments.

We could not include this in the main text because of space constraints, but have now developed this in detail in the Supplementary Discussion.

It would be better to compare/discuss the conservation and difference for SPO11 complex in different organisms, such as yeast, human, and mouse (maybe with the help of AlphaFold prediction?).

We have added a new figure (Extended Data Fig 12) that compares AlphaFold models of SPO11 complexes from different species.

Line numbers are required for review.

We apologize for this omission.

Page 4, about middle: "As expected", please clarify the rationale.

The rationale is implicit in the preceding sentence: "we tested whether phenol-chloroform partitioning leads to the enrichment of covalent protein-DNA adducts in the organic phase" (L89). We cannot add further details because of space constraints, but we added a cartoon in Fig. 2a to help clarify the idea.

Page 5, the first two paragraphs: Why prefer to bending and supercoiled DNA? Is this related to the observations that supercoiling affects DSBs and crossovers enriched in negative supercoiled regions in yeast (PMID: 36107772; PMID: 32152049)?

We don't know why SPO11 would prefer bendable and negatively supercoiled DNA. One possibility is that it is an evolutionary remnant of the substrate preference of Topo VI, imposed by structural constraints of the complex. Another possibility is that this property was evolutionarily conserved for its functional consequences, perhaps because it allows to link DSB formation to topological stress. A third possibility is that this is important mechanistically because the requirement for DNA bending prior to cleavage would constitute an activation barrier, and relaxation of the complex after cleavage could help drive the reaction forward.

We now included this in the Supplementary discussion (L725-731).

Page 6, "Dimerization controls DNA cleavage" and "SPO11 can reseal single-strand nicks": The data do not support these conclusions, please consider to revise them.

It is not clear from this comment what the reviewer objects to exactly. To the contrary, we think that both conclusions are amply supported by the data. In addition, they were also reached independently by the Keeney lab (accompanying paper by Zheng et al.).

Page 9, the last paragraph: Given a dimer is required for high cleavage activity, why is there only weak dimer interface?

That cleavage would require two SPO11 subunits was expected based on the molecular mechanism of type II topoisomerases, to which SPO11 is related. The central message of this manuscript is that the monomeric state of SPO11 in solution (i.e. the fact that it has a weak dimer interface) would serve as an effective mechanism to prevent unwanted DSBs. If dimerization *in vivo* is dependent on accessory partners, which themselves are spatially and temporally regulated by multiple mechanisms, this would allow to finely control when and where breaks are made. This model is discussed in L291-298 (main Discussion), L667-695 (Supplementary Discussion) and illustrated in Extended Data Fig. 12.

Fig. 1d,e: What is the top band in the gel? 1f, a doublet for nicked bands?

Figure 1d, the top band is a plasmid dimer, now indicated in the legend (L461).
Regarding the doublet in Figure 1f, please see the response to the comment below.

Fig. 2a: Supercoiled, nicked, and linear all look like doublets?

In some gels, we see some of the DNA species migrating as doublets. We think that this is an artefact caused by SDS. In the standard assay, the DNA is loaded on the gel in the presence of 1% SDS. We think that this affects the migration of DNA at the start of electrophoresis. Once the DNA and SDS are separated, the electrophoretic mobility might change, which could create this artefact. The effect was more evident in some experiments than others, but are never seen when DNA is ethanol precipitated before loading the gels (e.g. in phenol-partitioning experiments). We consider these points too minor to include in the manuscript, given the short format of the paper.

Referees' comments:

Referee #1:

The authors have done a good job addressing all comments. I support publication.

Referee #2:

The authors' revision has nicely addressed all of my scientific concerns and I fully support publication of this important paper in Nature. I still think that the title is both premature and a bit off topic--the major accomplishment here is the demonstration that Spo11 alone can form DSBs in vitro. But I leave that to the editors and authors to sort out.

We agree and suggest to use the following title instead: "SPO11 dimers are sufficient to catalyze DNA double-strand breaks in vitro".

Referee #3:

The authors have resolved my concerns. Congratulations.

Minor comments: The authors should add their interpretation to the doublets in Figures 1 and 2 into the legends.

We will do it if requested by the editors, but we think it is unnecessary detail.